



# Characterisation of the Manchester Aerosol Chamber facility

Yunqi Shao[1*], Yu Wang[1*], Mao Du[1*], Aristeidis Voliotis[1*], M.Rami Alfarra[1,2,‡] S. Fiona Turner[1,†] and Gordon McFiggans[1]

[1] Centre for Atmospheric Science, Department of Earth and Environmental Sciences, School of Natural Sciences, University of Manchester, Manchester, M13 9PL, United Kingdom
[2] National Centre for Atmospheric Science (NCAS), University of Manchester, Manchester, M13 9PL, United Kingdom

[‡] Now at Environment & Sustainability Center, Qatar Environment & Energy Research Institute, 34110, Doha, Qatar
[†] Now at AMETEK Land, Dronfield, Derbyshire, S18 1DJ, United Kingdom
*These authors all made equal contributions to this work.

*Correspondence to*: G. McFiggans (g.mcfiggans@manchester.ac.uk)

**Abstract.** This study describes the design of the Manchester Aerosol Chamber (MAC) and its comprehensive characterisation. The MAC is designed to investigate multi-phase chemistry and the evolution of aerosol physico-chemical properties from the real-world emissions (e.g. diesel engine, plants) or of secondary organic aerosol (SOA) produced from pure volatile organic compounds (VOCs). Additionally, the generated aerosol particles in MAC can be transferred to the Manchester Ice Cloud Chamber (MICC), which enables investigation of cloud formation in warm, mixed-phase and fully glaciated conditions (with T as low as -55 °C). MAC is an 18 m³ FEP Teflon chamber, with the potential to conduct experiments at controlled temperature (15-35 °C) and relative humidity (25-80 %) under simulated solar radiation or dark conditions. Detailed characterisations were conducted at common experimental conditions (25 °C, 50% RH) for actinometry and determination of background contamination, wall losses of gases ($NO_2$, $O_3$, and selected VOCs), aerosol particles at different sizes, auxiliary mechanism and aerosol formation. In addition, the influences of chamber contamination on the wall loss rate of gases and particles, and the photolysis of $NO_2$ were estimated.

## 1 Introduction

This manuscript provides a description and characterisation of a simulation chamber predominantly used for the investigation of the evolution of aerosol composition and properties resulting from coupled photochemistry and microphysics, and recently for the exposure of human subjects for investigation of human health impacts. Atmospheric aerosols have significant effects on air quality, regional to global climate, and human health (Lohmann and Feichter, 2005, Pope et al., 2002, Katsouyanni et al., 1997). Aerosol particles range from a few nanometers to several tens of micrometers in diameter. Their composition is complex, comprising inorganic and organic compounds, dependent on their sources which may be either primary (e.g. sea salt, dust, wildfires) or secondary, from the oxidation of gaseous precursors (Seinfeld and Pandis, 2016). Organic compounds contribute 20~90% of the mass of submicron aerosols in the Northern hemisphere (Zhang et al., 2007, Jimenez et al., 2009), and of an estimated 10000~100000 atmospheric organic compounds (Goldstein and Galbally, 2007), only around 10% have



been identified (Hallquist et al., 2009). Owing to this complexity, their chemical reaction pathways and properties lead to substantial outstanding challenges to the understanding of organic aerosol (OA) formation, transformation, fate and impacts (Hallquist et al., 2009). Such an inadequate understanding of aerosol particles, and particularly the organic fraction, leads to large uncertainties in understanding their role in air quality and global climate. Processes relating to organic-containing particles have consequently been a primary focus of studies in our chamber.

Over the last several decades, numerous field measurements have been conducted globally to characterise OA in the atmosphere (Gray et al., 1986, Hoffman and Duce, 1977, Turpin and Huntzicker, 1991, Turpin and Huntzicker, 1995, Hallquist et al., 2009, Jimenez et al., 2009, Zhang et al., 2007). However, isolation of chemical and microphysical processes from meteorology and other atmospheric processes can be challenging in ambient measurements (Becker, 2006). To better understand the sources, physicochemical properties and aging processes influencing atmospheric aerosols, simulation chamber facilities have been developed across the globe since the 1960s (Cocker et al., 2001b, Karl et al., 2004, Carter et al., 2005, Paulsen et al., 2005, Saathoff et al., 2009, Wang et al., 2011, Platt et al., 2013, Schnitzhofer et al., 2014, Wang et al., 2014, Leskinen et al., 2015, Babar et al., 2017, Gallimore et al., 2017, Leone et al., 1985). In principle, a simulation chamber is a controlled system to elucidate processes that occur in the real atmosphere (Barnes and Rudzinski, 2006), gas-phase reactions and chemical pathways (Carter and Lurmann, 1991, Seakins, 2010, Atkinson et al., 1992), SOA production (Hallquist et al., 2009, Carlton et al., 2009, McFiggans et al., 2019), new particle formation (Smith, 2016, Wang et al., 2020, Wagner et al., 2017, Dunne et al., 2016), cloud processes (Wang et al., 2011, Frey et al., 2018, Wagner et al., 2006), transformations and properties of real-world emissions (from vehicles; e.g. Liu et al. (2017), biomass burning; e.g. Hennigan et al. (2011), plants; e.g. Hohaus et al. (2016)) and health effect(Tong et al., 2018, Taylor et al., 2000).

The design of simulation chambers varies widely with respect to the light sources, chamber sizes, materials and operation conditions to address varied lines of research (Barnes and Rudzinski, 2006). The size of chamber facilities ranges from ~l to ~300 $m^3$ and are variously constructed from pyrex/quartz, aluminium, stainless steel and FEP Teflon. The light sources of chambers include artificial and natural solar radiation, leading to a convenient classification into indoor and outdoor chambers (Barnes and Rudzinski, 2006). Pyrex or quartz is widely used for chambers with a volume of less than 1 $m^3$, with a few larger, such as JPAC (1.45 $m^3$) (Ehn et al., 2014), Bayreuth chambers (2.4 $m^3$) (Behnke et al., 1988) and UAREC (>1 $m^3$) (Barnes et al., 1994). Owing to its reasonably inert nature and transparency towards short-wavelength lights, pyrex and quartz chambers enable ready access to radical generation studies. Also, pyrex and quartz chambers can enable temperature-dependent studies with the use of a cooling or heating bath. Metal chambers are usually built with a volume of 1~6 $m^3$, with exceptions such as AIDA chamber (85 $m^3$) (Wagner et al., 2006), MICC (10 m3) (Connolly et al., 2012) and CERN Cloud chamber (26





$m^3$)(Schnitzhofer et al., 2014). The largest advantage of the rigid metal chambers is the ability of experiments under varying

temperatures, enabling simulation of free tropospheric conditions (Wagner et al., 2006) and warm, mixed phase and fully glaciated clouds. FEP Teflon is widely used in medium to large chambers, such as FORTH-ASC (10 $m^3$)(Kostenidou et al., 2013), Manchester Aerosol Chamber (MAC) (18 $m^3$), LEAK-LACIS (19 $m^3$) (Mutzel et al., 2015, Niedermeier et al., 2020), IASC (27 $m^3$), Caltech dual chamber (28 $m^3$ and 90 $m^3$) (Cocker et al., 2001a, Carter et al., 2005), PSI chamber (27 $m^3$) (Paulsen et al., 2005), ILMARI (29 $m^3$) (Leskinen et al., 2015), HELIOS (90 $m^3$) (Ren et al., 2017), SAPHIR (270 $m^3$)(Karl

et al., 2004)  and EUPHORE ($2 \times 200$ $m^3$) (Bloss et al., 2005). The transparency of Teflon enables its widespread use in both indoor and outdoor chambers, enabling transmission across the solar spectrum.

All chambers have limitations. A universal challenge is the existence of chamber walls that can as a sink of the reacting gases and aerosol particles (McMurry and Grosjean, 1985) and a surface on which they can react. Consequently, experimental results relating to gas-particle partitioning, aerosol formation rate and yield, for example, require careful interpretation (McMurry and

Grosjean, 1985, Matsunaga and Ziemann, 2010, Zhang et al., 2014, Ye et al., 2016, Wang et al., 2018). Similarly, photochemistry experiments in indoor chambers using artificial lights requires consideration of the wavelength dependence of the irradiance (Barnes and Rudzinski, 2006). Outdoor chambers, particularly the larger ones, are challenged by control of relative humidity and temperature due to the ambient diurnal variation, hampering repeatability (Barnes and Rudzinski, 2006). Finally, the chamber background and its memory can affect repeatability and reliability of the results (Carter and Lurmann,

1991, Wang et al., 2011, Wang et al., 2014, Schnitzhofer et al., 2014). This necessitates a clear and detailed characterisation of chamber behaviour and history.

This paper provides a description of the design and the characterisation of a novel indoor simulation chamber, MAC, located at The University of Manchester. Equipped with state-of-the-science instruments, MAC has been used to explore the  aerosol

formation and aging (Hamilton et al., 2011, Alfarra et al., 2012), physicochemical properties of multi-component aerosol particles (Alfarra et al., 2013, Wang et al., 2021, Voliotis et al., 2021), gas-particle partitioning (Voliotis et al., 2021), aerosol formation, properties and transformations from plant emissions (Wyche et al., 2014) and from engine emissions (Pereira et al., 2018, Liu et al., 2017). Additionally, the entire contents of the MAC can be transferred directly to the MICC (Manchester Ice Cloud Chamber) to investigate the warm, mixed phase and fully glaciated cloud formation on the aerosol particles that will act

as cloud condensation nuclei (CCN) and ice nuclei (IN). A detailed description of the coupling between the facilities and its use can be found in Connolly et al. (2012) and Frey et al. (2018) and will not be discussed here.



## 2. Description of the MAC

The MAC is operated as a batch reactor where the composition of the gaseous precursors, pre-existing seed, oxidising environment, relative humidity and temperature are controlled throughout a typical experimental duration of several hours. It is equipped with a variable combination of gas-phase and particle-phase analytical instruments as listed in Table 1. The MAC consists of an 18 m$^3$ FEP Teflon bag suspended from a frame comprising a central fixed frame member and two moving members, all contained within a RH and temperature-controlled enclosure. Along with the light sources, cooling systems and air purification system, the MAC is shown in the schematic in Figure 1.

[Figure 1 about here.]

### 2.1 Enclosure and environmental control

The rectangular enclosure comprises an extruded aluminium framework supporting two access sides each with two bi-fold doors and two fixed sides within which the lamp enclosures and air conditioning (AC) ducts are situated. The inner walls and the ceiling of the enclosure and the floor are fully coated with reflective space blanket in order to approximate an integrating sphere to maximise the chamber irradiance and provide even light intensity. The temperature and relative humidity between the chamber and the enclosure walls is controlled by the AC allowing the temperature within the range of 15-35°C and RH between 25% and 80%. The inlet duct is positioned aloft at one end of the chamber and the outlet duct is at the bottom of the other such that conditioned air at 3 m$^3$s$^{-1}$ continually passes through the 50 cm space between the bag and enclosure, agitating and mixing the air in the bag as it does so. The RH setpoints is chosen to match the dewpoint of the chamber air at the desired temperature. Temperature and dew point is measured at two points in the chamber (at the middle and on the side) using a dew point hygrometer and two thermocouples to choose the setpoint.

### 2.2 Teflon reactor

The reactor comprises four sections of FEP Teflon film (50μm, AdTech Polymer Engineering Ltd.). FEP Teflon film is chosen since it is chemically inert and more transparent than PVF and PTFE, having better light transmission between 290 and 800nm, and has lower rates of hydrocarbon off-gassing (Finlayson-Pitts and James N. Pitts, 2000). A weakness of the FEP film is the



accumulation of electrostatic charge which can significantly increase the wall loss rates for particles with a diameter smaller than 500nm (McMurry and Rader, 1985, Charan et al., 2018).

The chamber is suspended in the enclosure and the Teflon films are compression-sealed and clamped and mounted on three pairs of rectangular extruded aluminium frames. The central rigid pair is fixed, with the upper and lower pairs free to move vertically. They are counter-weighted to enable the bag to expand and collapse when sample air is introduced and extracted in the process of fill/flush cycles and sampling. This maintains a very slight positive pressure, minimising contamination from

125 laboratory air. The top and bottom Teflon film sections are around 0.5m below and above the ceiling and floor of the enclosure respectively, when the bag is full of air. The central fixed frame pair supports 3 inlet and sampling manifolds constructed of solid Teflon, one in each of the two long sides and one in the short side of the bag, as well as mirrors and optical fibre mounting for a 2-pass broadband Differential Optical Absorption Spectroscopy (DOAS) system for retrieval of aerosol optical properties along the long axis of the chamber. One manifold is connected to the air purification system (described in 2.4) for injection of

130 purified air, VOC precursor, $NO_x$, $O_3$, seed aerosols and transfer of sample to selected instrumentation in the upper-floor laboratory. A second manifold is used for sampling gas and particulate material from the chamber to online instrumentation next to the chamber over the course of each experiment. A second port in this manifold can be used to couple the chamber to emission sources such as engines, plant chambers etc., and has been discussed elsewhere (Wyche et al., 2014, Pereira et al., 2018). The third manifold houses sensors to monitor the RH and T inside the chamber and at the chamber walls.

## 2.3 Chamber illumination

The irradiation source, consisting of two xenon arc lamps and a bank of halogen bulbs, is mounted inside the enclosure and is used to approximate the atmospheric actinic spectrum. Two 6 kW arc Xenon lamps (XBO 6000 W/HSLA OFR, Osram) are installed on the bottom-left and the top-right of the chamber housing, respectively. Quartz plates with optical polish (PI-KEM

Ltd) of 4mm thickness in front of each arc lamp filter out unwanted UV light. The absorption of IR light due to the water vapour in the atmosphere is simulated in the chamber by the usage of a container, equipped with quartz glass on either side and filled with ultra-high purity water, located in front of each arc lamp. The bank of 112 halogen lights, 7 rows of 16 bulbs each (Solux 50W/4700K, Solux MR16, USA), are mounted on the same enclosure wall as the bottom xenon arc lamp, facing the inlet.


The unwanted heat generated from the irradiation source is removed by the cooling system which includes the Air Conditioning (AC) unit, a chiller connected to the bottom arc lamp, and a recirculating water system connected to the top arc xenon lamp and running through aluminium bars cooling the halogen bulb holders. The chiller water circulates through tanks in front of each arc lamp faced by the quartz filter plates, dissipating heat produced by absorption of unwanted IR.

## 2.4 Chamber air purification, conditioning and injection system

Purified dry air is supplied by passing laboratory air at up to 3 $m^3$ $min^{-1}$ using a 3-phase blower (Nash Elmo; model G200) through a drier (ML180, Munters) and three filters; first a canister containing Purafil/charcoal, second, one containing activated charcoal, and third a Hepa filter to remove the $NO_x$, volatile organic compounds and particles. The clean air can be conditioned by passing through the humidifier, ozoniser and aerosol mixing tank before entering the chamber. The ozoniser (OZV30, Waterth) generates ozone using 2 mercury lamps. The custom-built humidifier comprises a 50L tank fed with ultra-pure water, producing water vapour using an immersion heater. VOCs are added to the chamber by injecting the desired liquid amount into a gently heated glass bulb (to ~80 °C) and transferred using the ECD grade nitrogen (N4.8; purity 99.998%; $N_2$) as the carrier gas. $NO_x$ (NO and $NO_2$) is added to the chamber using custom-made cylinders at 10% v/v and a mass flow controller and transferred with ECD grade $N_2$ as the carrier gas. Seed particles are generated by an atomiser (Topaz model ATM 230) and pass through a 0.12 $m^3$ stainless steel aerosol residence chamber before being flushed into the chamber. All components are connected with large bore (50 mm) stainless steel pipes apart from the diversion lines for the seed, humidifier and ozoniser, which have a 25 mm bore. The flow path is controlled by several 2- and 3-way electro-pneumatic valves along the inlet system. In general, the chamber maintains a relatively clean environment with particle concentrations <15 particles $cm^{-3}$, particle mass concentration ~0μg $m^{-3}$, $O_3$ concentrations ~0ppb and $NO_x$ concentration <10ppb (NO <8ppb and $NO_2$<2ppb).

## 2.5 Control system

To regulate the chamber operational procedures and devices (fill/flush cycles, injection procedure, humidification, VOC bulb heating, ozoniser operation) conveniently, repeatably and precisely, a bespoke automated control system is used. All component switches are controlled from a home-built PLC board, with all control signals processed using ladder logic and communicating with a graphical front end in Visual Basic. All components (including the 2- and 3-way valves) shown in Fig. 1 are controlled by the PLC. Selection of the valve position controls whether clean air is injected into the chamber, or the chamber contents are flushed to exhaust. Cycles of filling and flushing are programmed to enable unsupervised





operation during cleaning cycles. The humidifier and ozoniser can be bypassed by controlling the diversion line valves

during the fill part of the cycle. Relative humidity and temperature of the chamber were continuously measured via the Edgetech and Sensirion sensor that are also PLC controlled. All control data are saved automatically. Three pre-programmed operations (pre-experiment, post-experiment and fill/flush cycle) are provided in the control system to enable manual or automated operation. More details about these operational procedures are provided in section 2.7.

## 2.6 Modes of operation


The MAC generally operates as a batch reactor that provides a closed system without the continuous flow of reactants or dilution flow of clean air. There have been several modes of operation used. The most straightforward mode is the heated bulb injection of commercial pure VOC precursors to investigate SOA formation and transformation either in sole or mixed VOC systems (Hamilton et al., 2011, Jenkin et al., 2012). The MAC has additionally been coupled to whole combustion process and

biogenic emission sources. A dynamometer, diesel engine and oxidising catalyst unit can be connected to the chamber directly, allowing controlled exhaust dilutions by controlled injection timing of exhaust fumes into the chamber under selected loads and speeds. For example, Pereira et al. (2018) reported the effect of different engine conditions and emission control devices on unregulated diesel exhaust gas emissions. The MAC has been coupled with a custom-built plant chamber to investigate the SOA formation from the real plants under controlled conditions. Wyche et al. (2014) deployed the chamber to investigate SOA

formation from biogenic VOC precursors emitted from the silver birch and three South-east Asian tropical plant species. Also, the MAC infrastructure was recently successfully extended to continuously generate $NO_3$ radicals using synthesised $N_2O_5$, to enable studies of SOA formation and transformation under night-time conditions.

## 2.7 Experimental procedures

To ensure reliable and reproducible control of experimental conditions, three specific experimental procedures have been programmed to be sequenced and implemented automatically or manually to ensure a lower chamber background. The first, designated the pre-experiment procedure, includes several fill/flush cycles of the chamber with clean air at a high flow rate of $3 \ m^3 \ min^{-1}$ for ~1.5 h. The VOC glass bulb is also cleaned in the course of the pre-experiment procedure using ECD grade $N_2$.

The second, conducted at the end of each experiment, is the post-experiment procedure. Again, this consists of several fill/flush cycles of the chamber with clean air at a high flow rate of $3 \ m^3 \ min^{-1}$ for ~1.5 h, with a subsequently fill with a high





concentration of $O_3$ (~1ppm) to soak the chamber overnight to oxidize the residual $O_3$-reacting volatile species. A more aggressive "harsh cleaning" procedure is carried out weekly during experimental campaigns. In this procedure a high concentration of $O_3$ (~1ppm) is filled into the chamber with illumination, undergoing several hours of photooxidation at high
relative humidity (~80%).

## 2.8 Instrumentation

A range of instruments can be used to measure the physical and chemical properties of the chamber air, as shown in Table 1. The table is separated into two parts, displaying the core instrumentation which are permanently fixed at the chamber as well
as the additional instrumentation which can be coupled to the chamber and used on demand. All the instruments sampling from a number of ports in the manifolds, equipped with stainless steel or PTFE tubing extending to the middle of the chamber.

NO and $NO_2$ are measured using a $NO_x$ Thermo 42i chemiluminescence analyzer. $O_3$ is measured by an Thermo 49C analyser. Both $NO_x$ and $O_3$ analysers are regularly calibrated using certified cylinders and an ozone calibrator, respectively. Water-based
condensation particle counters (wCPC; model 3785 and 3786) have been selected as core instrumentation operating in the chamber room to avoid the interference of the volatile working fluid (e.g., butanol), usually found on other CPC units, to diffuse into the chamber. A wCPC is being used to measure the total particle number concentration in the chamber and the other is coupled to a differential mobility analyser (DMA, Brechtel Inc) as part of a custom-built differential mobility particle sizer (DMPS) system to measure particle size distributions in the 40-600 nm range. The DMA has been built in such a way,
so it uses filtered chamber air as sheath flow to maintain the gas-particle equilibrium during the measurements.

The chamber is equipped with a 47mm filter holder which is located at the flushing line of the chamber (see Fig. 1), enabling the sampling of the whole chamber contents at the end of each experiment at high flow rate ($3m^3$ $min^{-1}$) onto the desired substrate. In such a way, adequate amounts of particulate mass can be collected for subsequent off-line analysis.
A selection of additional instrumentation able to measure gas and particle phase composition and properties are available to be used on demand. Briefly, oxygenated VOCs are measured using a high-resolution time-of-flight chemical ionisation mass spectrometer (CIMS; Aerodyne/Tofware) using iodide as a reagent ion. Non-refractory $PM_1$ composition is measured using a high-resolution time-of-flight aerosol mass spectrometer (HR-AMS; Aerodyne) while oxygenated particulate organic
composition is measured using the filter inlet for gases and aerosols (FIGAERO) when coupled to CIMS. Total organic and





elemental carbon concentrations are measured using a semi-continuous carbon aerosol analyser (OC/EC; Sunset Laboratory; Model 4). Selection of particles based on their mass, or their aerodynamic size can be achieved using a centrifugal particle mass analyser (Combustion) and an aerodynamic aerosol classifier (Combustion), respectively. Particle hygroscopicity and volatility are measured by custom-built hygroscopicity tandem differential mobility analyser (HTDMA) and thermal denuder

(TD), respectively, while cloud condensation nuclei (CCN) activity is measured by a CCN counter (Droplet measurement Tech). Black carbon concentration and properties can be measured by a three-wavelength photoacoustic spectrometer and single particle soot photometer (Droplet measurement Tech).

Routinely, additional instrumentation, such as gas chromatograph coupled to a mass spectrometer (GC-MS) and proton transfer

reaction ionisation scheme (PTR) were added to MAC as part of collaborative work to measure VOCs concentrations (Alfarra et al., 2013, Wyche et al., 2014, Wyche et al., 2015). Similarly, particle offline analysis using liquid chromatography-mass spectrometry/tandem MS (LC-MS/MS) and two-dimensional GC-MS (2D-GC-MS) have been also employed occasionally to probe the chemical characteristics of the SOA particles (Hamilton et al., 2011, Wyche et al., 2015) .

[Table 1 about here.]

### 3. MAC Characterization

This section describes the characterisation of each element of the chamber with relevance to the operation and influence on the and interpretation of the experimental results.

### 3.1. Temperature and relative humidity

In the MAC, the temperature between the enclosure and the chamber is controlled by the AC system and of the chamber itself, by the temperature of air in this space. The calibrated dewpoint hygrometer (Edgetech sensor) is used as a reference for Sensirion capacitance sensors during dark conditions (no irradiation of the chamber) where there is no influence of temperature gradient caused by the light sources. The dark experiment at a set temperature of 16℃ and two photooxidation experiments at different set points of relative humidity (40% and 70%) and a set temperature of 25℃ were conducted to examine the

temperature and relative humidity homogeneity of the chamber.



Figure 2a shows the evolution of temperature at the chamber wall and at the centre of the chamber measured by the Edgetech and one Sensirion sensor, respectively, during dark and photooxidation experiments. The temperature accuracy of sensors is ± 0.3℃ and ± 0.2℃ for the Sensirion sensor and Edgetech sensor at 25℃, respectively. In photooxidation experiments, the

temperature in the chamber centre (24 ± 1℃) is stable and slightly higher than that in the chamber wall (23 ± 1℃). Such a gradient might be caused by the cooler air between the chamber wall and enclosure and incomplete mixing. The temperature in dark conditions shows good agreement with the two sensors, around 16℃.

Figure 2b displays the relative humidity results of the two photooxidation experiments measured by the Sensirion and Edgetech

sensors. The RH in the centre of the chamber measured by the Sensirion capacitance sensor (40 ± 1% and 70 ± 1%) was slightly lower than the RH at the wall of the chamber measured by the Edgetech hygrometer (39 ± 1% and 65 ± 1%)

[Figure 2 about here.]

**3.2. Mixing**

NO, $NO_2$ and $NO_x$ are selected as gas tracers to test the gas-phase mixing time inside the reactor. There are no fans or other equipment inside the chamber, however, $NO_x$ is injected as $NO_2$ into the 3 $m^3$ $min^{-1}$ (=50 L $s^{-1}$) flow through the 50 mm diameter inlet at a velocity of approximately 25 m/s, inducing near instantaneous mixing throughout the chamber. Throughout the duration of an experiment, the forceful agitation of the Teflon walls by the AC flow between the enclosure and chamber

continuously maintains mixing inside the reactor. As shown in Figure 3a, the mixing time for NO, $NO_2$ and NOx gases is somewhere less than 2 minutes. Typically, the mixing time in atmospheric simulation chambers fall in the range of minutes, for example, 1 min in the CESAM chamber with 4.2 $m^3$ (Wang et al., 2011) and 2 mins in the GIG-CAS chamber 30 $m^3$ (Wang et al., 2014)

Neutral seed particles (ammonium sulfate, AS) were chosen to examine the mixing time of particles in the chamber. Briefly, seed particles were injected into the seed aerosol residence chamber and mixed for 1min and subsequently introduced into chamber at the flowrate of 3 $m^3$ $min^{-1}$. Figure 3b shows the concentration of total particles measured by the wCPC as a function of time, which shows that the mixing time for seed particles in the chamber is around 2.5 mins. This time is comparable with the gases mixing time in the chamber.






[Figure 3 about here.]

### 3.3. Light intensity

The artificial radiation in the MAC has a broad radiation distribution owing to the chosen combination of illumination sources, producing irradiation over the wavelength range 290-800 nm to capture all wavelengths of the atmospheric actinic spectrum. Figure 4 shows the total actinic flux measured in MAC (red line) multiplied by 3.5 compared with the Manchester midday clear sky measurements on a June day.

[Figure 4 about here.]

The photolysis rate of $NO_2$ ($jNO_2$) estimated in steady-state actinometry can be used as a confirmation of the light intensity in the chamber (Hu et al., 2014) measured by direct spectral radiometry. Such actinometric measurements were carried out by injecting $NO_2$ into the chamber and irradiating for several hours, measuring the concentration of NO, $NO_2$ and $O_3$ continuously. A series of $NO_2$ actinometry experiments were conducted with ~ 70ppb $NO_2$ injected into the chamber and irradiated for more than 3 hours, with the temperature and humidity maintained at around 25°C and 50% respectively. The photolysis frequency of $NO_2$ is calculated from

$$jNO_2 = \frac{k_{NO+O_3} \times [NO] \times [O_3]}{[NO_2]} \tag{1}$$

where $k_{NO+O3}$ is the rate constant of the reaction of $O_3$ and NO ($1.8 \times 10^{-14}$ $cm^3 molecule^{-1} s^{-1}$ at 298K) (Atkinson et al., 2004).

In the MAC, we find the $jNO_2$ to be $2.25 \pm 0.4 \times 10^{-3}$ $s^{-1}$ from the actinometry experiments. This compares with a value of $1.5 \times 10^{-3}$ $s^{-1}$ from the integrated absorption across the measured wavelengths using the absorption cross-section and quantum yield from Daumont et al. (1992). This suggests that the $jNO_2$ from the actinometry experiments has reasonable agreement with the spectral radiometry given that the actinometry $jNO_2$ is an average and chemically-derived across the bag contents while the spectral radiometry is a point-measurement in an imperfect integrating sphere. It is also worth to noting that the $jNO_2$ of ambient midday in clear sky on June was $7 \times 10^{-3}$ $s^{-1}$, which 4.7 times greater than spectra radiometry. Our measured $jNO_2$ is generally comparable with those obtained across the broader simulation chamber community, as shown in Table 2.



Using the measured actinic flux and integrating across all wavelengths $jO(^1D)$, the photolysis frequency of ozone to yield $O(^1D)$, is found to be $1.23 \times 10^{-5}$ $s^{-1}$. This is slightly lower compared to the ambient $jO(^1D)$ ($1.5 \times 10^{-5}$ $s^{-1}$) owing to light

intensity reduction via UV filter. The ambient midday clear sky midsummer ratio of $jNO_2 / jO(^1D)$ is 467 and compares with our chamber value of 122.

### 3.4. Wall loss of gaseous compounds

Chamber wall adsorption had been shown to be a substantial source of gas losses inside Teflon bags (Wang et al., 2011) and

will influence the gas-phase reactivity and SOA formation. In the MAC, the wall loss rates of $NO_2$, $O_3$ and several volatile organic compounds ($\alpha$-pinene, toluene, 1.3.5-TMB and limonene) were investigated by injecting known concentrations into the chamber and measuring their concentration decay for an extended period under dark conditions. Approximately 50 ppb concentration of $NO_2$ and different concentration of $O_3$ (120-350 ppb) were injected into the chamber and monitored for 4 hours allowing sufficient time for a measurable decay. Selecting different concentration of $O_3$ could assist in investigating the impact

their initial concentration on the wall loss rate in the MAC. For the wall loss experiments of volatile organic species, 50 ppb of each compound was injected into the chamber with the decay monitored for ~4 hours. All the wall loss experiments of gaseous species were conducted under the T and RH of ~25°C and ~50% respectively.

The wall loss rates of gaseous species were calculated by considering their decay as a first-order process. The wall loss rates

of $NO_2$ and $O_3$ were $9.40 \pm 7.38 \times 10^{-7} s^{-1}$ and $2.09 \pm 0.97 \times 10^{-6} s^{-1}$, respectively. Table 2 compares the wall loss rates of $NO_2$ and $O_3$ between MAC and other chambers. The $NO_2$ decay rate at the MAC was slightly higher than all the other chambers listed, except GIG-CAS. Meanwhile, the $O_3$ decay rate of MAC was higher than TU and KNU chamber, but lower than GIG-CAS and PSI chamber.

The first order wall loss rate of the selected anthropogenic and biogenic VOCs were $2.24 \pm 0.67 \times 10^{-5} s^{-1}$ for $\alpha$-pinene, $2.08 \pm 0.54 \times 10^{-5} s^{-1}$ for limonene, $2.06 \pm 1.25 \times 10^{-5} s^{-1}$ for toluene and $12.22 \pm 0.90 \times 10^{-5}$ $s^{-1}$ for 1.3.5 TMB.

[Table 2 about here.]

[Figure 5 about here.]



### 3.5. Wall losses of particles

Particles are deposited to chamber walls mainly due to natural convection, diffusion, gravitational settling and electrostatic forces in addition to physical mixing (Crump et al., 2007, Pierce et al., 2008, McMurry and Rader, 1985). Several different

approaches have been proposed to account for these losses in the literature. Most commonly, AS seed total number and/or mass decay in the dark is used (Zhang et al., 2007, Murphy et al., 2006).

A series of experiments were conducted to investigate the size-resolved particle lifetimes under various humidity and mixing conditions using AS seed, which was introduced to the chamber and left in the dark at the desired RH and temperature

conditions for ≥4 hours. An initial seed concentration of 50-100 $\mu g$ $m^{-3}$ was used with a modal diameter of ~100 nm. The size-resolved concentration of the AS seed was monitored using a DMPS at 40-600 nm range, with a 10 min scanning time. Here, in line with the literature (Cocker et al., 2001a, Gallimore et al., 2017, Smith et al., 2019, Donahue et al., 2012), the particle wall-loss rate was retrieved by fitting an exponential function to the total mass and number concentrations, as well as in each size bin of the DMPS, to obtain a size-resolved decay rate coefficient. A comparison between the application of different

particle wall-loss correction methods is shown in section 3.5.1.

The mean ($\pm$ 1$\sigma$) number and mass wall loss rates were estimated as 9.17 $\pm$ 1.3 and 8.16$\pm$ 1.5 $\times$ $10^{-5}$ $s^{-1}$, respectively. Those values are comparable with those reported in the literature for chambers of various volumes, spanning from ~5 to 270 $m^3$ as shown in Table 3 (Cocker et al., 2001a, Gallimore et al., 2017, Rollins et al., 2009, Wang et al., 2014, Donahue et al., 2012). Although it may be expected the large differences in the chamber volumes to affect the particle wall losses, the surface to

volume ratio (S/V) of all the chambers listed on Table 4 are on the same order of magnitude. Roughly, the particle losses to the chamber walls show an increasing trend with decreasing S/V, possibly indicating that S/V can be determinant for the losses of particles.

The mean ($\pm$ 1$\sigma$) size-resolved wall loss rate ($s^{-1}$) of particles in the MAC at various relative humidity and mixing conditions

are shown in Figure 5. The size-resolved particle wall loss rate in all experimental types shows a decreasing trend with particle size. In the size range measured, such behaviour has been observed previously in chambers with varying volumes (Wang et al., 2018) and is a result of the high diffusivity of the particles in the sub-100nm range, in addition to a possible contribution of coagulation when particle number concentrations of the small size particles are high. The size-resolved decay rate values obtained here are comparable to those reported for the CMU chambers (Wang et al., 2018). The continuous agitation of the

chamber walls due to the AC of the enclosure affects the particle wall losses, with the experiments conducted under such conditions showing higher wall-loss rates compared to those where the AC was disabled. This behaviour may be related to





enhanced electrostatic losses (McMurry and Rader, 1985). The amount of the water vapour also affects the particle wall-loss rate with the experiments conducted under drier conditions having lower loss rates compared to those at moderate RH conditions. These results suggest that the experimental conditions can have a significant impact on the particle wall-loss rates.

Therefore, care should be taken when using the retrieved wall-loss rates from such experiments to correct the SOA particle mass in experiments conducted on different environmental conditions.

[Table 3 about here.]


### 3.5.1. Investigation of various wall-loss correction methods to the SOA formation

As mentioned above, the retrieved total and/or size-resolved ammonium sulfate particle loss rates from characterisation experiments are commonly used to correct the SOA particle mass from VOC oxidation experiments (Ng et al., 2007, Fry et al., 2014, Nah et al., 2017). Several alternative numerical approaches have also been proposed (Wang et al., 2018, Pierce et

al., 2008). Here, we compare three different approaches to correct the SOA particle mass from a β-caryophyllene photooxidation and a limonene ozonolysis experiment. More specifically, we use the size-resolved and the total mass loss rates retrieved from the characterisation experiments (described in Section 3.5.) as well as the modelling approach proposed by Verheggen and Mozurkewich (2006). The last approach employs inverse modelling to simulate the particle wall losses based on diffusion and gravitational settling, while the losses due to coagulation are indirectly inferred. We further correct the SOA

mass using the obtained ammonium sulfate wall-loss rates from the nearest in date characterisation experiment as well as the average values from experiments that conducted over 2 years. The results are summarised in Figure 6. It should be noted that our aim here is not to investigate the characteristics of each method, rather to demonstrate their effect when correcting for particle wall-losses in atmospheric simulation chambers.

[Figure 6 about here.]

The different approaches clearly result in substantially different wall-loss corrected SOA masses (Fig. 6). In all cases, the correction using the ammonium sulfate size-resolved wall-loss rates resulted in greater differences compared to the remaining two approaches. Moreover, the choice of the experiment used for ammonium sulfate loss rates can have a critical impact on

the resulted wall-loss corrected SOA masses. In our case, using the average ammonium sulfate loss rates resulted in





substantially-unrealistically higher SOA mass predictions compared to when the ammonium sulfate loss rates from the nearest characterisation experiment were used. Consequently, for the MAC, it is clearly recommended to conduct frequent wall loss characterisation experiments to enable more reliable correction.


### 3.6. Auxiliary mechanism

An auxiliary mechanism aims to describe the chamber wall reactivity such that it can be directly used as data set in future computer modelling to simulate the chamber experiments. A set of experiments were conducted including simulating clean air, dark decay of $NO_2$ and $O_3$. Four non-elementary hypothetical reactions and relevant parameters used in the model are listed

in Table 4. The parameters for the $NO_2$ and $O_3$ formation rate from the Teflon walls were calculated based on the off-gassing experiments under light irradiation conditions for at least 3 hours reaction. The initial concentrations of $NO_2$ and $O_3$ in the chamber were varied from 0 to 8 ppb. The light-induced formation of $NO_2$ and $O_3$ from the chamber walls were $6.95 \pm 1.26 \times 10^{-5} s^{-1}$ and $8.56 \pm 2.58 \times 10^{-5} s^{-1}$, respectively.

The decrease of $NO_2$ and $O_3$ in the gaseous phase under dark conditions could be explained as chamber wall adsorption as mentioned in the wall loss of gaseous compound section (section 3.4) for the new chamber bag, and discussion about usage and condition of the chamber's bag impact on the wall loss rate present in section 5.

[Table 4 about here.]

### 4. α -pinene photooxidation experiment

To evaluate the chamber facility for the purposes of studying SOA production and transformation, α-pinene photochemistry experiments were conducted in the MAC. The initial experimental conditions are shown in Table 5. During the experiments, chemical composition ($NH_4$, $SO_4$, $NO_3$, OA) in the particle phase and α-pinene in the gas phase were monitored by HR-ToF-AMS and Semi-continuous GC-MS, respectively. The measured SOA mass by HR-ToF-AMS was corrected due to the non-

unit collection efficiency of the instrument following standard procedures in previous studies (Jimenez, 2003, Jayne et al., 2000, Allan et al., 2003, Allan et al., 2004) and chamber wall loss effects (Wang et al., 2018).





To compare with literature data, SOA yield (Y) was used as a proxy to evaluate SOA production (Grosjean and Seinfeld, 1989), defined as the SOA mass formation ($\Delta M_o$) from the reactive organic gas ($\Delta VOC$) consumption as shown in Eq. (2).

Here, the SOA mass is wall loss corrected using the size-resolved wall-loss rate of ammonium sulphate particles from the nearest characterisation experiment as described in Section. 3.5.1. Odum et al. (1996) incorporated gas/particle partitioning theory (Pankow, 1994a, Pankow, 1994b) into SOA formation and calculated SOA yield from individual compounds, shown in Eq. (3). Here, $Y_i$ represents the yield of compound i. $\alpha_i$ is a stoichiometric factor representing the ratio of the molecular weight of product i to the parent VOC. $K_{p,i}$ and $C_{OA}$ are the partitioning coefficient of product i and the total absorbing organic

mass (the same as $\Delta M_o$ herein). Furthermore, Odum et al. (1996) successfully used a two-product model parameterising SOA yield and $\Delta M_o$ as shown in Eq. (4). The $\alpha_1$, $\alpha_1$, $K_{p,1}$, $K_{p,2}$ can be fitted upon yield curves.

$$Y = \frac{\Delta M_0}{\Delta voc} \tag{2}$$

$$Y = \sum_i Y_i = C_{OA} \sum_i \left( \frac{\alpha_i K_{p,i}}{1 + K_{p,i} C_{OA}} \right) \tag{3}$$

$$Y = C_{OA} \left( \frac{\alpha_i K_{p,i}}{1 + K_{p,i} C_{OA}} + \frac{\alpha_i K_{p,i}}{1 + K_{p,i} C_{OA}} \right) \tag{4}$$

The yield curves as a function of $\Delta M_o$ for the three $\alpha$-pinene experiments in this study and the comparison with literature data (Cocker et al., 2001b, Saathoff et al., 2009, Eddingsaas et al., 2012, Stirnweis et al., 2017) are shown in Figure 7 (all yield curves are wall loss corrected). As expected from the absorptive partitioning, it can be seen that the SOA yield increased

consistently with an increase of absorptive organic mass for the three $\alpha$-pinene experiments in this study. Our results are qualitatively and quantitatively comparable with $\alpha$-pinene photochemistry experiments under different oxidant conditions and seed initialisations in other chambers, such as with HONO & $H_2O_2$ as oxidant and no seed & AS seed & acidic seed in Caltech chamber (Eddingsaas et al., 2012) and with VOC/$NO_x$ ratio in the range of 0.2 to 25 and AS/Ammonium bisulphate (ABS) seed in PSI chamber (Stirnweis et al., 2017). Additionally, we use the two-product model to fit the yield curve of three $\alpha$-

pinene experiments in this study and the fitted $\alpha_1$, $\alpha_1$, $K_{p,1}$, $K_{p,2}$ are 0.03, 0.34, 3.14e+006, and 0.02, respectively, as shown in Figure 7 (black solid line). The fitted yield curve for $\alpha$-pinene photochemistry with aqueous AS seed in this study is comparable to the $\alpha$-pinene ozonolysis without seed (Cocker et al., 2001b, Stirnweis et al., 2017), but much higher than the ozonolysis with aqueous seed (Cocker et al., 2001b).




[Table 5 about here.]

[Figure 7 about here.]

## 5. Effects of contamination on chamber performance


It has been shown that organic vapours can condense on the Teflon chamber walls in a similar manner to the losses of particles (Matsunaga and Ziemann, 2010, Zhang et al., 2014, Krechmer et al., 2020). The deposition of those compounds on the chamber walls can be reversible (Matsunaga and Ziemann, 2010) or quasi-irreversible (Ye et al., 2016) and proportional to each compound's volatility and chemical characteristics. Similarly, other atmospheric gases, such as HONO, have been also found to condense on chamber walls (Rohrer et al., 2005). The uptake of semi-volatile vapours from the chamber walls has been proven to substantially affect the oxidative chemistry and thereby the reported SOA formation potential (Zhang et al., 2014, Rohrer et al., 2005). It is therefore likely that both gaseous and particle deposition can lead to a build-up of contamination on chamber walls with time. It is not guaranteed that any cleaning procedures are completely effective, and it is important to consider the experimental history of a chamber when interpreting experimental behaviour (and particularly when comparing experiments conducted in different periods). To assess the effect of contamination from such sources on the chamber performance, we conducted the same characterisation experiments as those described in Sections 3.3-3.5, in an extensively used Teflon bag, after a series of experiments with high concentrations of particles and gases derived from diesel engines.



The photolysis rate of $NO_2$ (i.e., $jNO_2$) from actinometry was found to be lower in the extensively used bag compared to a newly installed bag ($1.83 \pm 0.47$ vs. $2.25 \pm 0.40 \times 10^{-3}$ s$^{-1}$, respectively). Assuming no physical change or degradation of the lamps installed in the MAC, the differences in the $jNO_2$ over the bag usage may reflect the changes in the chemical behaviour of the chamber film or in the wavelength dependent transparency of the walls. There was found a substantial increase in the wall loss rate of $NO_2$ and $O_3$; in the extensively used bag these were $7.95 \pm 6.90 \times 10^{-6}$ s$^{-1}$ and $2.23 \pm 1.83 \times 10^{-5}$ s$^{-1}$, respectively, compared to $0.93 \pm 0.76 \times 10^{-6}$ s$^{-1}$ and $0.20 \pm 0.08 \times 10^{-5}$ s$^{-1}$ in the newly installed bag, respectively. At the same time, the wall production (i.e., off-gassing) of the same gases was decreased (0.12 vs. $0.19 \pm 0.04 \times 10^{-7}$ s$^{-1}$ for the $NO_2$ and 0.20 vs. $0.24 \pm 0.07 \times 10^{-7}$ s$^{-1}$ for the $O_3$). Unfortunately, spectral radiometry data are not available for the extensively used bag, which could help to identify whether the reduction in the $jNO_2$ is attributed to the transparency of the walls over usage of the bag or the changes in the production and loss of gases from and to the chamber walls. Nonetheless, applying the average wall-loss rates of $NO_2$







and $O_3$ derived from the clean and used bag to the measured data during an actinometry experiment in the extensively used
bag results in $jNO_2$ values of 1.93 and $2.07 \times 10^{-3}$ s$^{-1}$, respectively. This suggests that correcting the gas data using the wall
loss rates obtained in the extensively used bag, shifts the $jNO_2$ value closer to the average for the clean bag (i.e., $2.25 \pm 0.4 \times 10^{-3}$ s$^{-1}$). Consequently, the changes in the wall loss rates of gases seem to be responsible for the calculated $jNO_2$. In addition
to the changes in the wall losses of gases, similar changes were observed in the wall losses of particles. The losses of the total
particle mass were $7.83 \pm 1.66 \times 10^{-5}$ s$^{-1}$ in the extensively used bag vs. $6.50 \pm 1.66 \times 10^{-5}$ s$^{-1}$ in the clean bag, implying that
the chamber usage can also affect the mass losses of particles. Conversely, losses of the total particle number appeared to be
unaffected by the history of the bag, exhibiting similar values in both conditions ($9.00 \pm 3.33$ vs. $9.00 \pm 1.66$ s$^{-1}$). Wang et al.,
(2018) reported significant changes in the wall-loss rates of particles after major maintenance activities in the area where the
chamber was suspended and attributed those differences to the electrostatic forces caused by friction. In our setup, the chamber
is enclosed in a housing and the operators have little to no contact with its walls, so it may be unlikely this to be the main cause
for the changes in the particle wall-losses over the bag usage history.

## 6. Discussion and conclusions

In this work, the MAC facility was comprehensively described and characterised. MAC is a batch reactor and showed good
temperature and relative homogeneity, parameters that can influence the SOA formation and partitioning  (Stirnweis et al.,
2017;Cocker et al., 2001b;Saathoff et al., 2009). Although our reported experiments in this study were performed only under
certain conditions, the results shown demonstrate that MAC can provide controlled temperature and relative humidity
conditions, which are important for any systematic chamber study. MAC however is limited to a RH range of 25-80% and
temperature of 15-35°C, owing to the heat generated by the lamps and the capacity of the AC unit.

Due to its explicit setup, the generated light spectrum mimics well the ambient solar radiation spectrum, comparable to that in
Manchester, yet having lower total actinic flux by a factor of ~3.5. Furthermore, fast mixing times are effected by the injection
of the reactants at high flow rates, while the air circulation around the chamber housing continuously agitates the chamber
walls resulting in sufficient mixing of its components during the experiments.

The bespoke control system of the MAC allows the generation of automated procedures that can enhance the comparability
across experiments. Moreover, due to its design, gases and particles generated via a number of sources can be introduced to


the chamber and studied in detail. In addition, its unique capability of transferring the whole contents of MAC to MICC provides the grounds for aerosol-cloud interaction studies (e.g., Frey et al., 2018).

Different wall loss rates of $NO_2$ were observed between MAC and other chambers, as shown in Section 3.4. Possibly, the wall

loss rates of gaseous compounds are affected by experimental conditions (such as temperature and RH) and chamber sizes (Metzger et al., 2008, Wang et al., 2011). Importantly, we showed that the usage history can influence the wall loss of gases with a higher wall loss rate of $NO_2$ and $O_3$, which may result in the lower $jNO_2$ in an extensively used bag, as showing in Section 5. Slightly higher wall particle mass loss rates were observed in an extensively used bag, while the particle number loss rates appeared to be unaffected. Wang et al., (2018) did not observe significant changes in the particle number loss rates

over 3-year undisturbed conditions in the CMU smog chamber. Meanwhile they showed that the particle number wall-loss rates can be substantially higher, as by factor of 3-4, due to electrostatic forces caused by friction. In our case, as shown in Section 5, it is more likely that the contaminated chamber walls may provide additional sinks to absorb more particles and gases irreversibly.

Additionally, the various methods for the particle wall-loss correction led to different wall-loss corrected SOA masses, which in turn can have substantial implications for the derived SOA yields (Odum et al., 1996, Wang et al., 2018, Hoffmann et al., 1997), as shown in Section 3.5.1. Furthermore, using the particle wall loss rate coefficients from experiments conducted at different times over the lifespan of a bag (e.g, nearest experiments, or all experiments), we showed that there might be a significant variation on the particle loss rates over the bag history. This illustrates that using different approaches or

experimental dataset to conduct such corrections may result in bias in the SOA yields and tahat the particle wall-loss correction is not a straightforward task in smog chambers (Stirnweis et al., 2017, Cocker et al., 2001b, Saathoff et al., 2009).

Our measured SOA yield curve from the photo-oxidation of α-pinene in the presence of seed particles, appeared to be comparable with other studies that conducted ozonolysis experiments in the absence of seed particles (Cocker et al., 2001a;

Stirnweis et al., 2017), but much higher than the ozonolysis with aqueous seed (Cocker et al., 2001b), as showing in Fig 7. However, it should be considered that the comparison of yield curves between different laboratories and facilities is quite complicated as there are many factors (seed/no seed, oxidants, relative humidity, VOC/NOx ratios, wall loss correction methods and etc,) that will affect the yields curves. Also, the characterization parameters of a chamber (e.g. gases and particles wall-loss rates) may also play an important role in the SOA formation as shown in Section 3.5.1. and discussed further above.



Furthermore, the loss of condensable vapours to the chamber walls can result in a lower SOA formation even for high seed concentration conditions (Zhang et al., 2014).

Based on our results, regular characterisation experiments are recommended in order to track chamber's performance while accounting for any potential changes to the interpretation of the results. Considering that the atmospheric simulation chambers

are composed by various materials and they come in different designs, sizes and shapes, in turn affecting their performance and behaviour, the comparability of their results should be a crucial priority of the scientific community. The results presented here highlight the need in developing a set of simple, standardised experiments and/or procedures that can be used from chambers across the globe in an effort to elucidate the characteristics of each facility and the interpretation of their results.


**Competing interests**

The authors declare that they have no conflict of interest.


**Data Availability**

The data that support the findings of this study are openly available in EUROCHAMP-2020 programme (https://data.eurochamp.org/data-access/chamber-experiments/).


**Author contributions**

GM, MRA, AV, YW, MD and YS conceived the study. GM, MRA and SFT designed the MAC. AV, YW, YS and MD conducted the experiments, the data analysis and wrote the manuscript with inputs from all authors.

**Acknowledgements**

The Manchester Aerosol Chamber was supported by the EUROCHAMP2020 research programme funded by the European Union's Horizon 2020 research and innovation programme under grant agreement no. 730997. AV and MD acknowledge the financial support from Presidents Doctoral Scholarship from the University of Manchester. AV acknowledges the support by the Natural Environment Research Council (NERC) EAO Doctoral Training Partnership. YW acknowledges CSC scholarship



support. MRA acknowledges funding support from the Natural Environment Research Council (NERC) through the UK National Centre for Atmospheric Science (NCAS). Instrumentational support was funded through the NERC Atmospheric Measurement and Observational Facility (AMOF).





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









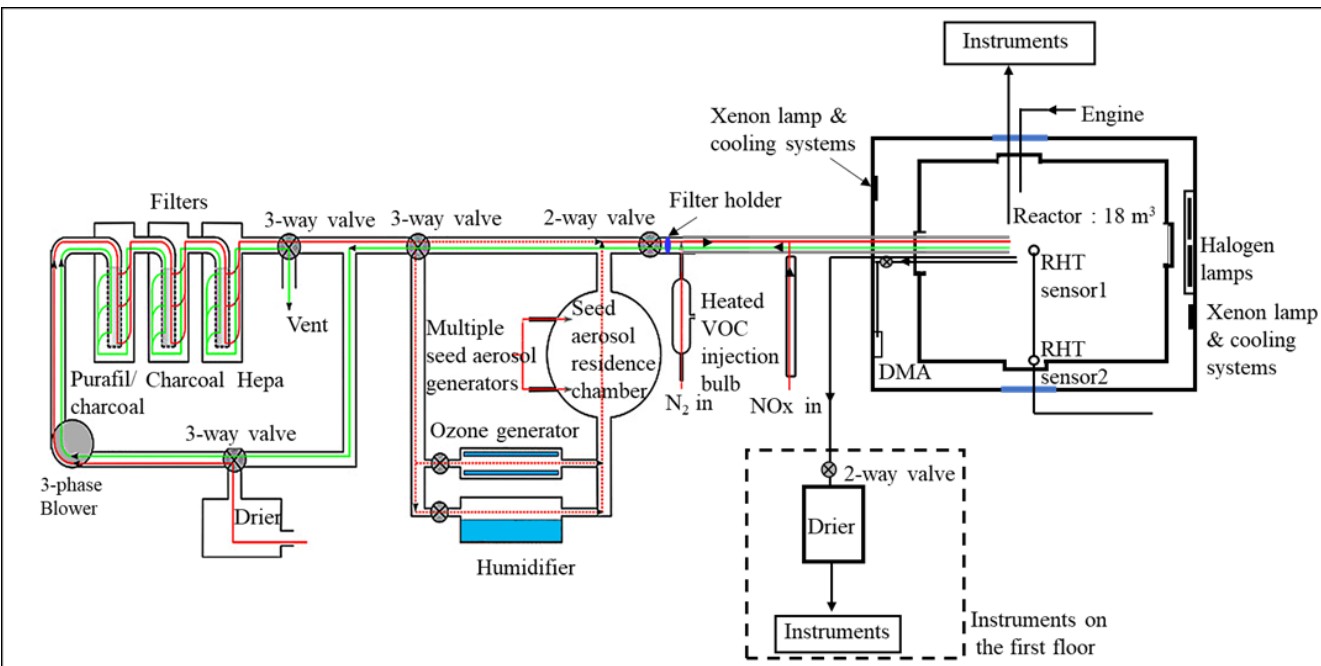

**Figure 1:Schematic of the Manchester aerosol chamber**





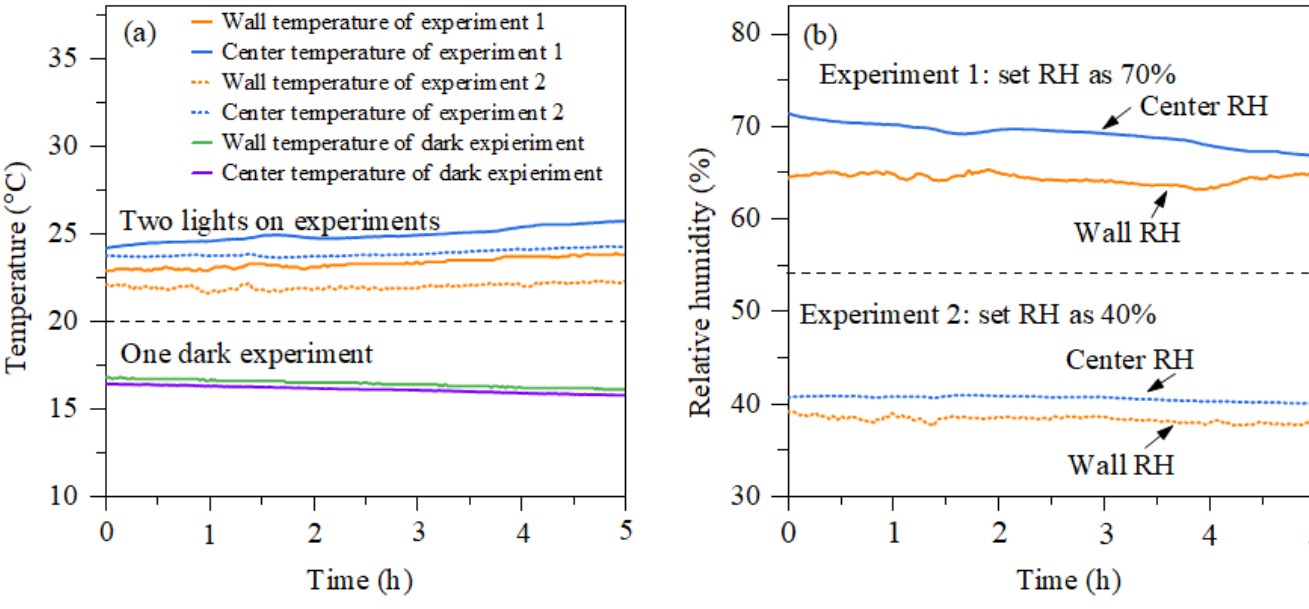


**Figure 2:(a) Temperature as a function of time measured by the Edgetech (wall temperature) and Sensirion (centre temperature) in a dark experiment and a photooxidation experiment. (b) RH as a function of time in the two photooxidation experiments.**





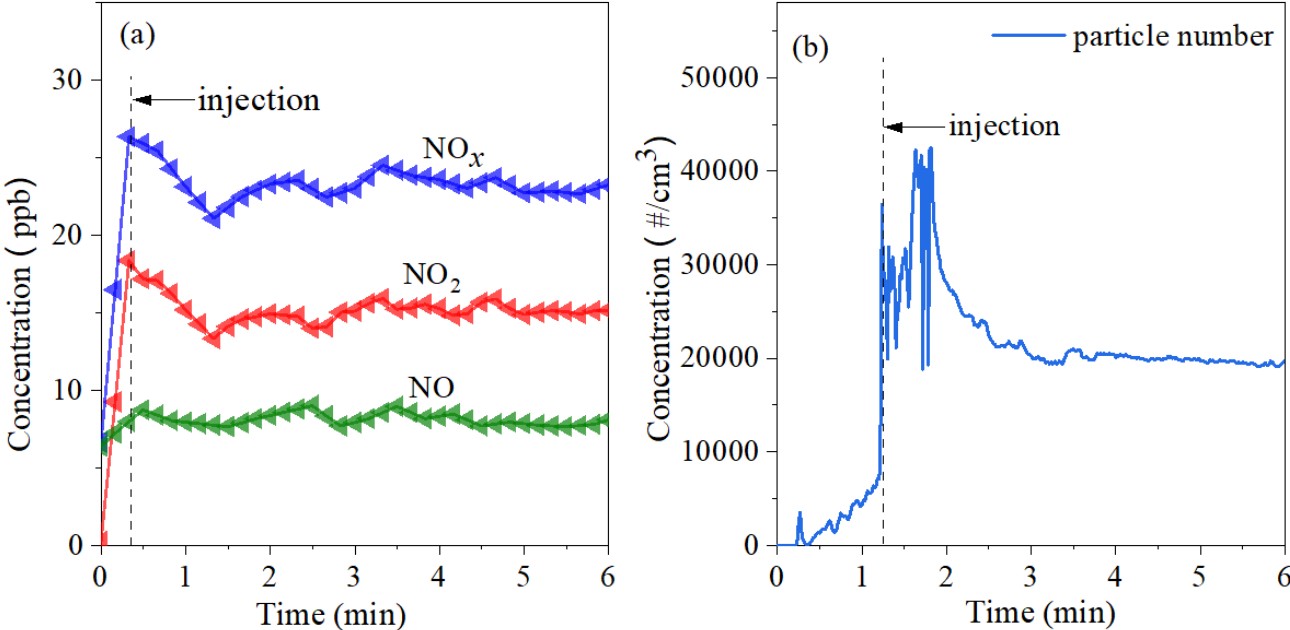

**Figure 3: (a) Gases ($NO_x$, $NO_2$ and NO) and (b) particle number concentrations as a function of time after their injection to the MAC.**



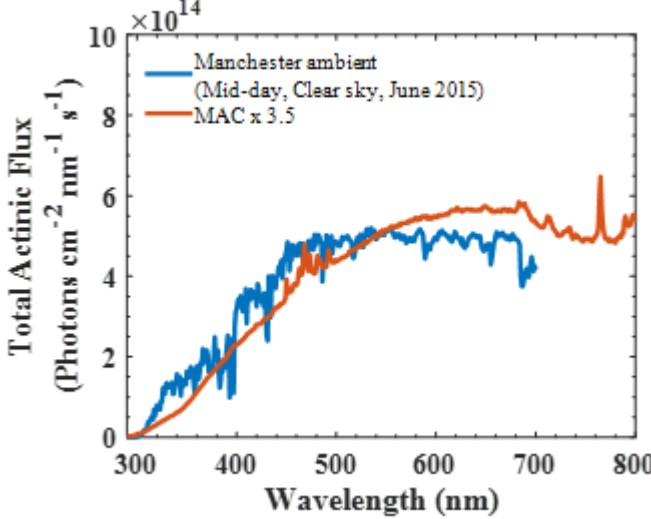

**Figure 4: Total actinic flux spectrum in the MAC compared to the ambient light spectrum obtained in the city of Manchester (UK) mid-day with a clear sky in June 2015.**






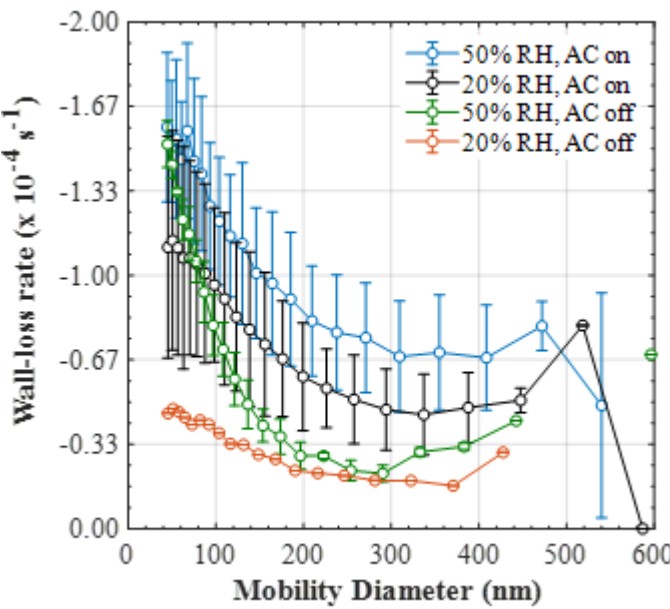

**Figure 5:Mean (± 1σ) size-resolved wall loss rate (s⁻¹) of particles in the MAC at various relative humidity and mixing conditions**
**(50% RH and mixing, n=9; 20% RH and mixing, n=5; 50% RH and no mixing, n=3; 20% RH and no mixing, n=1)**







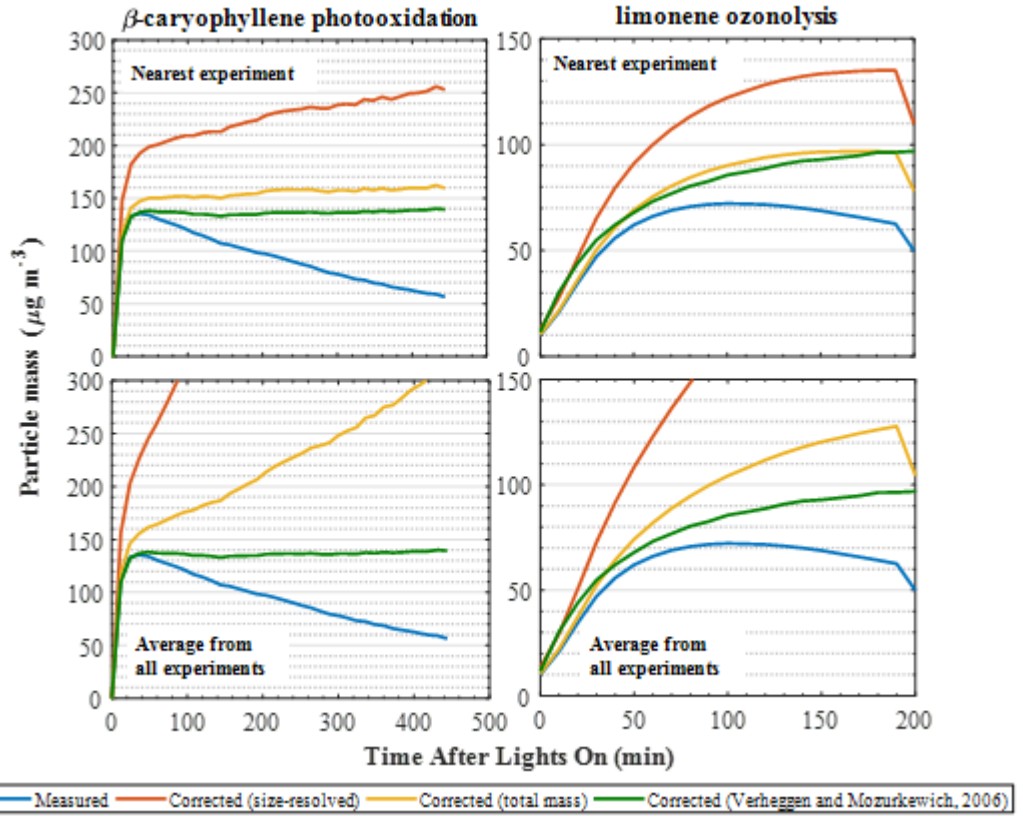

**Figure 6: Measured and wall-loss corrected SOA particle mass using three different wall-loss correction approaches for a β-caryophyllene photooxidation (left panels) and a limonene ozonolysis (right panels) experiment.**







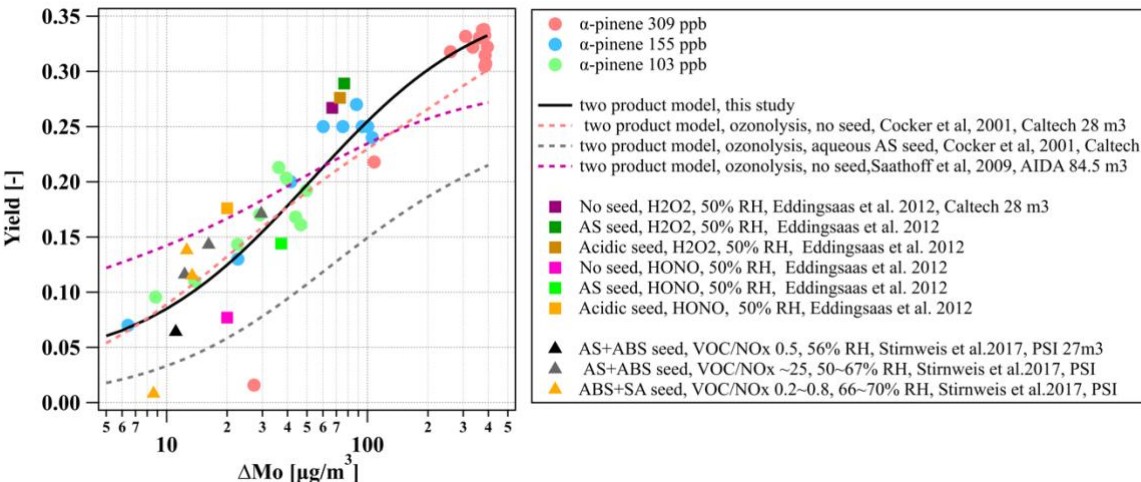


**Figure 7: Yield curve for α-pinene photochemistry on aqueous AS seed experiments in this study and literature data (Cocker et al., 2001b, Saathoff et al., 2009, Eddingsaas et al., 2012, Stirnweis et al., 2017). All experiments are carried out under humid conditions. Lines represent the two-product model fit for yield curves.**








**Table 1: List of the available instrumentation at MAC**

| Instrument | Model | Measured parameter | LOD/ range |
|---|---|---|---|
| *Core instrumentation* | | | |
| Dew point hygrometer | Edgetech; DM-C1-DS2-MH-13 | Dew point | -20 – 90 ± 0.2 ºC |
| Sensirion capacitance sensor | Sensirion; SHT75 | Temperature, relative humidity | -40 to +125, ±0.3 °C 0 – 100, ±1.8 % |
| NOx analyser | Thermo; 42i | NO, NO$_2$ | 0.5 to 1000 ppb |
| O$_3$ analyser | Thermo; 49C | O$_3$ | 0-0.05 to 200 ppm |
| CO analyser | Thermo;48i | CO | >0.04 ppm |
| Water-based condensation particle counter, wCPC | TSI; 3785, 3786 | Particle number | <$10^7$ p/cc |
| Differential mobility particle sizer | Custom-built[a] | Particle size | 40-600 nm |
| Filter collector | Custom-built[b] | Particle collection for offline analysis | |
| *Additional instrumentation* | | | |
| Condensation particle counter, CPC | TSI; 3776 | Particle number | <$10^7$ p/cc |
| Scanning mobility particle sizer, SMPS | TSI; 3081 | Particle size | 10-1000 nm |
| Aerodynamic Aerosol Classifier, AAC | Cambustion | Selection of particles by size | 25-5000 nm |
| Centrifugal Particle Mass Analyser | Cambustion | Selection of particles by mass | Mass accuracy: 5% |
| High-resolution aerosol mass spectrometer, HR-AMS | Aerodyne | PM$_1$ non-refractory particle composition | >0.05 µg m$^{-3}$ |
| Iodide chemical ionisation mass spectrometer, I$^-$-CIMS | Aerodyne/Tofware | Oxygenated VOC | LOD >60 ppt; Mass resolution 4000 Th/Th |
| Filter Inlet for Gases and AEROsols, FIGAERO | Aerodyne/Tofware | Particle composition | >$10^2$ ng |

| | | | |
|---|---|---|---|
| Semi-Continuous Carbon Aerosol Analyzer, OC/EC | Sunset Laboratory; Model 4 | Organic/elemental carbon concentration | >0.5 µgC m⁻³ |
| Hygroscopicity tandem differential mobility analyser, HTDMA | Custom-built [c] | Hygroscopicity | 20-350 nm |
| Cloud condensation nuclei counter, CCNc | Droplet measurement Tech; CCN-100) | CCN activity | $>6 \times 10^3$ particles cm⁻³ at SS:0.2% |
| Thermal denuder | Custom-built[d] | Volatility | Temperature range: ambient – 200°C |
| Three-wavelength photoacoustic spectrometer, PAS | Droplet measurement Tech | BC | 0-100,000Mm⁻¹ |
| Single Particle Soot Photometer, SP2 | Droplet measurement Tech | light absorbing property of soot | >10 ng m⁻³ |

[a](Alfarra et al., 2012)  [b](Hamilton et al., 2011)  [c](Good et al., 2010)  [d](Voliotis et al., 2021)





**Table 2: Comparison of wall loss rate of NO₂ and O₃, jNO₂ with other chambers**

| Chamber | Wall loss rate (s⁻¹) | | jNO₂ (x10⁻³ s⁻¹) | Reference |
|---|---|---|---|---|
| | NO₂ | O₃ | | |
| MAC | $0.94 \times 10^{-6}$ | $2.09 \times 10^{-6}$ | 2.25 | This study |
| KNU | $7.45 \times 10^{-7}$ | $1.08 \times 10^{-6}$ | 2.83 | Babar et al., 2016 |
| GIG-CAS | $2.32 \times 10^{-6}$ | $2.18 \times 10^{-6}$ | 8.17 | Wang et al., 2014 |
| PSI | $2.17 \times 10^{-7}$ | $4.00 \times 10^{-6}$ | - | Metzger et al., 2008 |
| TU | $6.95 \times 10^{-7}$ | $1.02 \times 10^{-8}$ | 3.83 | Wu et al., 2007 |
| UCR | - | - | 3.17 | Carter et al., 2005 |
| AIOFM-CAS | - | - | 3.50 | Hu et al., 2014 |




**Table 3: Comparison of particle number wall loss rates with other chambers**

| Chamber | Particle wall loss rate (× $10^{-5}$ $s^{-1}$) | Chamber volume ($m^3$) | Chamber surface to volume ($m^{-1}$) | Reference |
|---|---|---|---|---|
| MAC | 9.17 | 18 | 2.33 | This study |
| KNU | 6.67 | 7 | 3.15 | Babar et al., 2016 |
| GIG-CAS | 4.67 | 30 | 5.13 | Wang et al., 2014 |
| Caltech | 5.50 | 28 | 1.99 | Cocker et al., 2001a |
| CMU | 11.00 | 12 | n.a. | Donahue et al., 2012 |
| SAPHIR | 7.50 | 270 | 2.68 | Rollins et al., 2009 |
| CASC | 5.67 | 5.4 | 3.44 | Gallimore et al., 2017 |
| NCAT | 15.83 | 9 | 2.90 | Smith et al., 2019 |




**Table 4: Auxiliary mechanisms and rates for chamber-dependent reactions. [a]Production rate of gaseous species from wall under light condition (P w, l). [b]Loss rate of gaseous species to wall under dark condition (L w,d).**

| Parameters | Gas Species | Rate (mean $\pm 1\sigma$) /s$^{-1}$ | Experiment |
|---|---|---|---|
| (P w,l)[a] | $NO_2$ | $(6.95 \pm 1.26) \times 10^{-5}$ | Direct measurement of $NO_2$ wall production |
| | $O_3$ | $(8.56 \pm 2.58) \times 10^{-5}$ | Direct measurement of $NO_2$ wall production |
| (L w,d)[b] | $NO_2$ | $(9.40 \pm 7.39) \times 10^{-7}$ | Direct measurement of $NO_2$ wall loss |
| | $O_3$ | $(2.09 \pm 0.9) \times 10^{-6}$ | Direct measurement of $O_3$ wall loss |





**Table 5:Summary of initial conditions for α-pinene photochemistry experiments**

| Exp. Date | VOC type | $[VOC]_0$ (ppbV) | VOC/NO$_x$ | T (°C) | RH (%) | AS Seed conc. (ug/m$^3$)[a] |
|---|---|---|---|---|---|---|
| 28-Mar-2019 | α-pinene | 309 | 7.7 | 26.7 | 50.5 | 60.7 |
| 6-Jul-2019 | α-pinene | 155 | 6.0 | 25.9 | 53.1 | 61.3 |
| 13-Jul-2019 | α-pinene | 103 | 5.7 | 27.2 | 54.5 | 55.4 |

[a]measured NR-PM mass concentration by HR-ToF-AMS with corrected collection efficiency (30 mins average before lights on).