# Peer review of "Characterisation of the Manchester Aerosol Chamber facility"

_Atmospheric Measurement Techniques, 2021_

## Author Comment (AC1)

**Response to comments of anonymous referees # 1**

This manuscript focuses on the description and characterization of the MAC chamber. It contains a detailed description of the chamber itself and results from different tests that were performed such as loss rate of trace gases and degassing from the chamber walls, loss rate of particles, etc.

I think the paper is relatively well written and it covers a good amount of testing. One thing is not clear to me though is if the chamber is a new chamber or the improvement of an older chamber or neither. I think it is not a new chamber as there are references of studies done with it since 2012. Therefore, I think it would be useful as this paper summarize the chamber itself to also summarize all the studies done within this chamber and in particular it would be interested, if possible, to show together in this paper the results of previous characterizations. One of the message of this study is that the chamber needs to be properly checked routinely during experiments as it is shown that indeed the behavior of the chamber in respect of degassing and losses of gases and particles changes significantly. Therefore I assume there are more tests available that could help track the behavior of the chamber over a longer time. This could be the paper where such information could be included.

If the chamber is new and/or if it consists of a drastic change from the previously used chamber it should be more precisely mentioned as it is not clear as it is described at the moment.

I also think it could be interesting to include within this paper the standardized set of tests that should be performed to properly characterized the chamber before new experiments are carried out.

We thank the reviewer for their time and effort in providing comments for our manuscript. Indeed, our chamber is not newly built and has been widely developed since 2005, in order to understand the chemical and physical properties of aerosols from different sources (e.g. engine, real-plant emission, biogenic or anthropogenic VOC). In all cases, a particular focus has been on the evolution of the changing composition and properties of the gaseous and particle constituents as they undergo simulated atmospheric processing. There has hitherto been less regard for quantification of the yield of components that has necessitated full quantification of the "confounding" processes associated with walls and mixing effects, for example. Nonetheless, aspects of these have previously been reported where necessary. We have taken advantage of our recent exploration of SOA formation from mixed precursors to perform a more systematic characterisation than previously attempted. These quantifications supersede our previous, more ad hoc, investigations of such processes.

A clearer statement of the chamber status has been added at the start of the last paragraph:

*"This manuscript provides a description and characterisation of an indoor simulation chamber, Manchester Aerosol Chamber (MAC), located at The University of Manchester. Although the MAC has been developed and predominantly used since 2005, the majority of the preceding studies were aimed to understand the chemical and physical properties of aerosols from different sources. There has therefore been less regard for quantification of the yield of components that has necessitated full quantification of the "confounding" processes associated with walls and mixing effects, for example. Here, we present a more systematic characterisation than previously attempted, along with a detailed description of the unique features of the MAC compared to other atmospheric simulation chambers"*

Specific comments:

Introduction. I feel the first sentence belongs more to the end of the introduction rather than to the beginning.

We agree with the reviewer. This sentence has been moved to the last paragraph of introduction.

Page 2, line 34. What does their stands for? I assume organic compounds but it could be specified.

Indeed this was referring to organic compounds. This has been now specified as suggested:

"..an estimated 10000~100000 atmospheric organic compounds (Goldstein and Galbally, 2007), only around 10% have been identified, such as alkanes, carbonyls, alcohols, easters, acids and etc (Hallquist et al., 2009, Goldstein and Galbally, 2007)"

Page 2, line 37. It would be good to have a reference at the end of sentence.

The reference has been added:

"*Such an inadequate understanding of aerosol particles, and particularly the organic fraction, leads to large uncertainties in understanding their role in air quality and global climate (McFiggans et al., 2006).*"

Page 2, line 48. I am sure there are more recent studies out there where chamber studies helped elucidating gas-phase reactions and chemical pathways (see everything that was done on isoprene and Criegee intermediates as two examples...).

Indeed, there are quite recent studies elucidating the gas phase chemistry and to reflect that we have added some additional references as:

"*...gas-phase reactions and chemical pathways (Carter and Lurmann, 1991, Seakins, 2010, Atkinson et al., 1992, Surratt et al., 2010, Paulot et al., 2009, Ehn et al.,2012, Bianchi et., 2019, Thornton et al., 2020),*"

Page 3, line 72. I would rephrase "A universal challenge is the presence of walls that can be a sink of the ..."

We agree with the reviewer. The sentence has been rephrased according to their suggestion.

Page 3, Lines77 and 78. Although it is true that humidity and temperature cannot be controlled in large outdoor chamber in the same way they can be in smaller indoor chamber, I find the sentence relatively misleading. First, at least for gas-phase mechanisms development, a difference of 20% in water content in a chamber will not affect the reproducibility of a study and the water content can be

reproduce better than that. Same goes for temperature which although of course having a larger effect, it is unlikely that two consecutive day of measurement in an outdoor chamber will see striking differences in the temperature in the chamber and experiments can be repeated when the conditions are similar. So, I would recommend taming down the sentence a little bit as to avoid giving the impression outdoor chambers might not give reproducible results.

We fully agree with the reviewer and we appreciate that our original sentence might have been misleading. The sentence has now been rephrased.

"*Outdoor chambers, particularly the larger ones, are challenged by control of relative humidity and temperature due to the ambient diurnal variation, which may introduce some challenges in the interpretation of the results (Barnes and Rudzinski, 2006)*"

Figure 1. I found it a bit hard to follow from figure one what was described in the text. I think adding more information (labels) in the figure would help the reader and, if available, I think a real picture of the chamber would help the reader following the discussion of the different parts of the chamber.

Thanks for both reviewers. Figure 1 has been revised.

Page 6, line 153-154. After the semicolon I think the main verb is missing.

We thank the reviewer for pointing out this error. The sentence has been rephrased to:

"*Purified dry air is supplied by passing dried laboratory air at up to 3 $m^3$ $min^{-1}$ using a 3-phase blower (Nash Elmo; model G200), a drier (ML180, Munters) and three filter canisters; the first containing Purafil/charcoal, the second containing activated charcoal, and the third with a HEPA filter, to remove the NOx, volatile organic compounds and particles, respectively*"

Page 7, line 195. I cannot quite identify which one is the third specific experimental procedure.

Thanks to the reviewer for pointing this out. The third specific experimental procedure is the harsh cleaning described in line 214-217. This sentence has been rephrased to:

"*The third, a more aggressive "harsh cleaning" procedure is carried out weekly during experimental campaigns. In this procedure a high concentration of O3 (~1ppm) is filled into the chamber with illumination, undergoing several hours of photooxidation at high relative humidity (~80%).*"

Page 8, line 210. Sample instead of sampling.

Addressed.

Page 9, Lines 250-251. I find this sentence not very clear. I would recommend rephrasing it.

We agree that this sentence might have been confusing. This sentence has been rephrased to

"The temperature in MAC is controlled by the AC system, which compensate the releasing heat from illumination system"

Page 10, lines 266-266. Any possible explanation why the RH differed between center of the chamber and at the walls?

Indeed, as the reviewer points out, there is differences of RH between the centre of the chamber and at the walls. The possible cause for this variance might attributed by the radiative heating. The RH and T difference between the centre and the edge of the bag is larger in light experiments, compared to the dark experiment where the RH and T differences between the centre and edge of our chamber are falling within the measured variance, i.e., $40 \pm 1\%$ vs. $39 \pm 1\%$ RH and $24 \pm 1°C$ vs $23 \pm 1°C$ T.

To reflect this, in the revised manuscript we had altered the text in section 3.1 that reads as:

"Figure 4 shows the temperature and relative humidity, measured at the edge (Edgetech sensor) and at the middle of the MAC (Sensirion sensor) for three characteristic experiments; one conducted in the dark and two in the presence of light. In the light experiments, it appears that both the temperature and humidity were higher in the centre of the MAC than that of the wall, while in the dark experiments these differences were negligible as they were within the uncertainty of our measurement. A likely explanation for this unexpected behaviour in the light experiments can be possibly to the radiative heating of the sensors in these experiments that could result in an over-estimation of the RH."

Citation: https://doi.org/10.5194/amt-2021-147-RC1

**Reference**

Bianchi, F., Kurtén, T., Riva, M., Mohr, C., Rissanen, M. P., Roldin, P., Berndt, T., Crounse, J. D., Wennberg, P. O., Mentel, T. F., Wildt, J., Junninen, H., Jokinen, T., Kulmala, M., Worsnop, D. R., Thornton, J. A., Donahue, N., Kjaergaard, H. G., and Ehn, M.: Highly Oxygenated Organic Molecules (HOM) from Gas-Phase Autoxidation Involving Peroxy Radicals: A Key Contributor to Atmospheric Aerosol, Chemical Reviews, 119, 3472-3509, 10.1021/acs.chemrev.8b00395, 2019

Ehn, M., Kleist, E., Junninen, H., Petäjä, T., Lönn, G., Schobesberger, S., Dal Maso, M., Trimborn, A., Kulmala, M., Worsnop, D. R., Wahner, A., Wildt, J., and Mentel, T. F.: Gas phase formation of extremely oxidized pinene reaction products in chamber and ambient air, Atmos. Chem. Phys., 12, 5113-5127, 10.5194/acp-12-5113-2012, 2012

Goldstein, A. H. and Galbally, I. E.: Known and Unexplored Organic Constituents in the Earth's Atmosphere, Environmental Science & Technology, 41, 1514-1521, 2007

McFiggans, G., Artaxo, P., Baltensperger, U., Coe, H., Facchini, M. C., Feingold, G., Fuzzi, S., Gysel, M., Laaksonen, A., Lohmann, U., Mentel, T. F., Murphy, D. M., O'Dowd, C. D., Snider, J. R., and Weingartner, E.: The effect of physical and chemical aerosol properties on warm cloud droplet activation, Atmos. Chem. Phys., 6, 2593-2649, 10.5194/acp-6-2593-2006, 2006.

Paulot, F., Crounse John, D., Kjaergaard Henrik, G., Kürten, A., St. Clair Jason, M., Seinfeld John, H., and Wennberg Paul, O.: Unexpected Epoxide Formation in the Gas-Phase Photooxidation of Isoprene, Science, 325, 730-733, 10.1126/science.1172910, 2009.

Surratt, J. D., Chan, A. W. H., Eddingsaas, N. C., Chan, M., Loza, C. L., Kwan, A. J., Hersey, S. P., Flagan, R. C., Wennberg, P. O., and Seinfeld, J. H.: Reactive intermediates revealed in secondary organic aerosol formation from isoprene, Proceedings of the National Academy of Sciences, 107, 6640, 10.1073/pnas.0911114107, 2010.

Thornton, J. A., Shilling, J. E., Shrivastava, M., D'Ambro, E. L., Zawadowicz, M. A., and Liu, J.: A Near-Explicit Mechanistic Evaluation of Isoprene Photochemical Secondary Organic Aerosol Formation and Evolution: Simulations of Multiple Chamber Experiments with and without Added NOx, ACS Earth and Space Chemistry, 4, 1161-1181, 10.1021/acsearthspacechem.0c00118, 2020.

---

## Author Comment (AC2)

**Response to comments of anonymous referees # 2**

This paper describes an atmospheric chamber that has been in use since at least 2011, providing details that were promised in a paper published at that time. Numerous papers have been published based upon experiments performed using this chamber though from the few I have examined, details of the chamber design, and key operating characteristics such as the mixing and wall losses have not been reported, but reports from the Manchester group show much deeper and more rigorous understanding and modeling of those factors than is evident in this manuscript.

We thank the reviewer for their assertion that we have a deep and rigorous understanding of the key operating characteristics of our facility.

1) This chamber is similar to other Teflon chambers; one key difference is the connection between this chamber and a cloud chamber.

We fully agree with the reviewer that a particularly unique feature of our chamber is the integrated design of its coupling to our cloud chamber. This has been the focus of previous manuscripts and the full characterisation of the coupling is beyond the scope of the current paper, which provides a detailed characterization of the MAC.

2) The chamber can also be interfaced to emission sources such as engines, as has been done in a number of chambers around the world. The manuscript mentions, but does not discuss these features, so this work needs to be evaluated in the context of reports on other atmospheric chambers, and in the context of the prior work reported from this chamber.

Again, we thank the reviewer for pointing out the additional valuable applications of our chamber (such as coupling to combustion sources and plant chambers). The behaviour of the coupling to emission sources is indeed important. However, this behaviour is limited to bespoke tailored experiments and does not impact upon the performance or behaviour of the chamber when conducting the more "standard" investigations of coupled (photo)chemistry, aerosol microphysics and the evolution of particle composition and properties. This characterisation is sufficiently challenging that we choose not to overextend the study. The behaviour of the coupling systems in these "real emission" applications are covered in the relevant publications and are also out of scope of the current paper.

3) While the design of atmospheric chambers is certainly within the purview of AMT, and the analysis of mixing and wall effects of a facility that has been used in numerous studies of fundamental atmospheric processes would be of value in understanding the results from those and future experiments, this paper offers little in the way of new insights.

MAC has been developed since 2005, predominantly to understand the chemical and physical properties of aerosols from different sources (e.g. oxidation of biogenic or anthropogenic VOCs, diesel exhaust, real-plant emissions,) and to relate them to one another. In all cases, a particular focus has been on the evolution of the changing composition and properties of the gaseous and particle constituents as they undergo simulated atmospheric processing. There has hitherto been less regard for quantification of the yield of components that has necessitated full quantification of the "confounding" processes associated with walls and mixing effects, for example. Nonetheless, aspects of these have been reported where necessary. We have taken advantage of our recent exploration of SOA formation from mixed precursors to perform a

more systematic characterisation than previously attempted. These quantifications supersede our previous, more ad hoc, investigations of such processes.

4) Careful selection of key experiments from the many that have been performed with this system, and more critical analysis and discussion of those results in light of current understanding of nonidealities of Teflon chambers could well change that assessment.

As detailed in our response above, a more comprehensive historical characterisation of our facility does not exist. However, we have simultaneously developed a capability to model chamber processes as recognised by the reviewer. Such chamber modelling accounts for all relevant wall interactions with particles and vapours, though constraint of such aspects remains challenging. We have provided an example of the capabilities of our chamber model to inform our understanding of the behaviour of Teflon chamber walls, though definitive constraint is an ongoing activity that will require careful dedicated experimentation (see responses #32).

5) Instrumentation has changed dramatically since earlier reports on chamber characteristics, e.g., Carter et al. 2005, and Cocker et al., 2001, so with the array of instruments listed in Table 1 it may be possible to provide data on wall effects beyond what has been previously reported using existing data from this chamber.

Indeed, the availability of advanced instrumentation over the past several years has the potential to provide more insightful information on the wall effects compared to studies conducted over a decade ago (e.g., Cocker et al., 2001; Carter et al., 2005). Such instrumentation was largely unavailable to us during our chamber characterisation experiments. Instruments such as the chemical ionisation mass spectrometer (CIMS) that are listed under the "additional instrumentation" section of the table is shared "non-core" equipment that is available for specified project periods, not continuously. A comprehensive investigation of wall effects will require bespoke experiments. As the PyCHAM model is maturing sufficiently to enable robust linkage between evolution of species adsorbed to the walls and those within the chamber, future CIMS measurements will target species identified by simulations as being particularly sensitive to wall interactions. Similarly, the rates of change of particle number distributions required to provide optimum model-measurement agreement will be interrogated to establish whether our understanding of particle wall losses is reliable.

6) Unfortunately, this paper repeats observations that have been reported many times, and that are commonly addressed in analysis of data from chamber experiments, albeit with a different chamber.

This is a key point. Whilst such observations have been reported for other chambers, the measurements made in this paper have not previously been published and the geometry, mode of operation, means of mixing and other design aspects will impact on the characterisation (as the reviewer points out throughout). We contend that it has not been suggested previously that all Teflon chambers behave identically in all regards and our characterisation presents as valid a contribution to the discussion of Teflon chamber behaviour as many others. Please also see the response to the next point.

7) One may ask what aspects of this chamber can be considered unique. It is a Teflon chamber operated in batch mode like many others. The air cleaning system is similar to other facilities. The chamber suspension system is a bit different from most Teflon reactors. One difference is the illumination system, which appears to more closely approximate the ambient

actic spectrum than many other facilities. Another is the automated cleaning system, which possibly could reduce run-to-run variability.

This is the other side of the coin addressed in the previous comment. Our chamber, whilst having many common features with others constructed of Teflon, has a number of features recognised by the reviewer. We thank the reviewer for this acknowledgement and concede that we could have more adequately emphasised their importance.

An unusual design feature of the MAC is the chamber suspension system, aiming to bestow several advantages in terms of duty cycle and efficiency improvement. The suspension system is intrinsically coupled into the chamber control, ensuring repeatability of the state of the chamber at the commencement of each experiment and consistency in geometry (and consequent surface to volume ratio) throughout experiments. The intention is to have more control over the geometry during collapse and expansion, rather than allowing a fixed bag to assume its natural geometry and to relieve stresses imposed on the structure in so-doing. We have no fixed structure for comparison, so have no direct evidence to prove show that run-to-run variability is improved, but we suspect it to be. We respond to the intended advantages relating to the automated cleaning facilitated by our suspension and control system in point 8 below and also to the bespoke illumination configuration in point 20.

We expanded the description in section 2.2, as follows:

*"The chamber is suspended in the enclosure and joints between three pairs of edges of the Teflon film are made by compression-sealing between the three pairs of rectangular extruded aluminium frames. The edges of the top and bottom Teflon webs are clamped by stainless steel clips installed on the aluminium frames with expanded foam strips relieving between the frame and Teflon to ensure even compression between the Teflon sheets. This approach avoids additional contamination from glue or tapes. The central rigid frame is fixed, with the upper and lower frames free to move vertically. They are counter-weighted to enable the bag to expand and collapse when sample air is introduced and extracted in the process of fill/flush cycles and sampling. This reduces the possibility of the chamber operating under negative pressure, minimising instrumental sampling problems and contamination from laboratory air. In normal practice, around 80% of the chamber air can be extracted from the chamber within ~ 5 min at a flow rate of 3 $m^3$ $min^{-1}$ in each flush cycle, after that the purified air can be filled into the chamber at the same flowrate. A low background condition is achieved in around 2-hours of continuous automated fill/flush cycles. This relatively rapid cleaning improves the duty cycle and efficiency of the chamber preparation process."*

8) Experimental impacts of the latter items are not discussed in this paper.

In order to illustrate the performance of the automated cleaning and filling system, Figure 1 below shows the gas and particle concentrations before and after their injection to the MAC from three identical experiments (mean ± 1σ). As can be observed from panels a and b, during the cleaning cycle the mixing ratios of NOx and O3 are sharply decreasing from ~40 and ~500 ppb, respectively, down to <10 and <1 ppb, respectively, during our automated filling cycle in less than an hour. Similarly, the particle number and mass concentration decreases down to <10 particles $cm^{-3}$ and 0 $\mu gm^{-3}$, respectively, prior to the injection of the reactants to the

chamber (Fig. 1c and e), so does the mixing ratio of a selected VOC (α-pinene) to 0 ppb (Fig. 1d). Furthermore, after ~3h of illumination in our cleaned bag (i.e., clean air + light experiments) the particle number and mass concentrations remain at the background levels, as indicated by Figure 2 below. This shows that our overall chamber gas-phase background is sufficiently low to prevent the formation of particles in the presence of lights and in the absence of reactants. Overall, before the addition of the reactants to the MAC, our automated cleaning procedure ensures rapid cleaning that results in repeatably low background concentrations

Fig. 1 and Fig. 2 below are now included in the revised version of the manuscript while the following text has been added to section 2.7:

"*These procedures ensure a clean environment is provided in the MAC prior to SOA experiments. Gaseous and particle time series before and after injection of reactants in three α-pinene photooxidation experiments in the presence of AS seeds are shown in Figure 2. As can be observed from panels a and b, during the cleaning cycle the mixing ratios of NOx and $O_3$ are sharply decreasing from ~40 and ~500 ppb, respectively, down to <10 and <1 ppb, respectively, during our automated filling cycle in less than an hour. Similarly, the particle number and mass concentration decreases down to <10 particles $cm^{-3}$ and 0 $\mu gm^{-3}$, respectively, prior the injection of the reactants to chamber (Fig. 2c and e), so does the mixing ratio of a selected VOC (α-pinene) to 0 ppb (Fig. 2d). Furthermore, after ~3h of illumination in our cleaned bag (i.e., clean air + light experiments) the particle number and mass concentrations remain at the background levels, as indicated by Figure 3. This shows that our overall chamber gas-phase background is sufficiently low to prevent the formation of particles in the presence of lights and in the absence of reactants. Overall, before the addition of the reactants to the MAC, our automated cleaning procedure ensures rapid cleaning that results in repeatably low background concentrations.*"

[Figure]

**Figure 1:** Time series (mean ± 1σ; n=3) of $O_3$ (a), $NO_x$ (b), particle number (c), VOC (α-pinene; d) and particle mass (e) from three identical experiments conducted in the MAC. Annotations provide information of the related process occurring in the chamber at each time point, normalised to the injection time of the reactants. Cleaning process duration is ~2h, while subsequently the chamber is left with clean air for about 1h prior the addition of the reactants ("clean chamber"). After the addition of the reactants, the chamber is stabilised in the dark for another hour ("dark stabilization") before the lights are turned on.

[Figure]

**Figure 2.** Time series (mean ± 1σ; n=2) of particle number (a) and mass (b) concentrations over the duration of air+lights experiments.

9) The mixing time in the chamber is fast, a couple of minutes. This is the primary feature of the MAC chamber. No demonstration, or even discussion is provided regarding novel experiments that might be enabled by this fast mixing time.

We thank the reviewer for acknowledging one of the features of our chamber. Indeed, our rapid mixing times can, for example, facilitate perturbation experiments beyond the standard "lights on/lights-off" experiments with the addition of reactants with near instantaneous mixing and reactions. Such experiments are in the agenda of our group in order to take advantage of techniques such as factor analysis (PMF and ME2) that rely on temporal variation on species behaviour.

10) Wall agitation is suggested to "result in sufficient mixing of its components during experiments," though no evidence is provided.

In order to provide more detailed evidence for the ability of our system to maintain sufficient mixing, we have replaced Fig. 3 of the original manuscript with a similar figure (Fig. 3 below) that averages data from three identical experiments to further illustrate the repeatability of our system.

As can be seen, after the addition of NOx and seed aerosol to the chamber and the air condition is turned on, the mixing ratios of NOx attain a value within a few minutes of their injection that remains constant thereafter. Similarly, the number concentration of the seed aerosol shows some fluctuations over the first ~10 mins and then stabilises , such that the particles subsequently experience only their expected losses to the chamber walls. The stability in the measured concentrations of these tracers provide evidence for the effectiveness of the mixing of the components of the MAC, while the low standard deviations between the experiments (shown as shaded areas) further demonstrate the repeatability that can be achieved in our system.

[Figure]

**Figure 3.** Mean ($\pm$ 1$\sigma$; n=3) (a) gas mixing ratio of NOx (ppb) and (b) particle number concentrations (particle cm$^{-3}$) in three characteristic experiments as a function of their time after their injection to MAC.

11) Wall agitation by the air conditioning will also enhance losses, which is demonstrated. The control system that automates filling and cleaning processes could, as suggested in the discussion and conclusions, enhance comparability across experiments. Again, no evidence is provided. As I read the paper this struck me as a potentially significant benefit; data to support the hypothesis would strengthen the paper.

Figures 1-3 and our responses to comments #8 and 10 highlight those points made by the reviewer. For example, Figures 1 and 2 demonstrate how our automated cleaning procedure can lead to repeatably low background concentrations of key species measured over a number of identical experiments. Similarly, Figure 3 expands on this by further showing the repeatability of our injections and the effectiveness of our mixing system. These results provide direct evidence on the ability of our system to reproduce similarly low background levels and comparable injections of reactants to the chamber.

12) The experiments that are discussed in the paper focus on wall effects, wall losses of particles and vapours, and wall reactions. A key conclusion is that wall losses make the chamber sensitive to effects of deposits during prior experiments, which is well known, and has been extensively explored by Huang et al. (Env. Sci. Tech. 2018). The authors speculate that increased $NO_2$ and $O_3$ losses with chamber age, and speculate that the losses may reduce $jNO_2$ (not explored by Huang), again without experimental evidence.

The experimentally derived apparent $jNO_2$ that was calculated based on photostationary state calculations was found lower in the "extensively used" compared to a "newly installed" bag. This reduction coincided with the increase of the $NO_2$ and $O_3$ wall loss rates and the decrease of their wall-loss production with the age of the bag (see L477-481.of the original manuscript). As we mention in the original manuscript (L481-483), in the absence of further experimental evidence, such as actinic flux data for both clean and dirty bag conditions, we can only speculate that the changes in the $jNO_2$ over the bag history could be attributable to the differences in the wall loss rates of these gases. As further mentioned in the response of the comment #5 future work will be focused in assessing the wall effects from the MAC by conducting dedicated experiments with advanced instrumentation. An alternative explanation could be that the transparency of the bag across the full wavelength range of our lamps is degraded by heavy use. This is recognised in our revised text.

In order to make the above discussion clearer in the revised version of the manuscript we have altered the wording of the Section 5, while we have included some information that derived from the study of Huang et al., 2018 that now reads as:

"*It has been shown that organic vapours can condense on the Teflon chamber walls in a similar manner to the losses of particles (Matsunaga and Ziemann, 2010, Zhang et al., 2014, Krechmer et al., 2020). The deposition of those compounds on the chamber walls can be reversible (Matsunaga and Ziemann, 2010) or quasi-irreversible (Ye et al., 2016) and proportional to each compound's volatility and chemical characteristics. Similarly, other atmospheric gases, such as HONO, have been also found to condense on chamber walls (Rohrer et al., 2005). The uptake of semi-volatile vapours from the chamber walls has been proven to substantially affect the oxidative chemistry and thereby the reported SOA formation potential (Zhang et al., 2014, Rohrer et al., 2005). It is therefore likely that both gaseous and particle deposition can lead to a build-up of contamination on chamber walls with time (Huang et al., 2018). It is not*

*guaranteed that any cleaning procedures are completely effective, and it is important to consider the experimental history of a chamber when interpreting experimental behaviour (and particularly when comparing experiments conducted in different periods). To assess the effect of contamination from such sources on the chamber performance, we conducted the same characterisation experiments as those described in Sections 3.3-3.5, in an extensively used Teflon bag, after a series of experiments with high concentrations of particles and gases derived from diesel engines.*

*The photolysis rate of NO2 (i.e., jNO2) derived from photostationary state calculations was found to be lower in the extensively used bag compared to a newly installed bag (1.83 ± 0.47 × 10-3 vs. 2.25 ± 0.40 × 10-3 s-1, respectively). This coincided with a substantial increase in the wall loss rate of NO2 and O3 that was observed in the extensively used compared to the newly installed (7.95 ± 6.90 × 10-6 s-1 vs 0.93 ± 0.76 × 10-6 s-1 for the NO2 and 2.23 ± 1.83 × 10-5 s-1 vs. 0.20 ± 0.08 × 10-5 s-1 for the O3, respectively). At the same time, the wall production (i.e., off-gassing) of the same gases was decreased (0.12 vs. 0.19 ± 0.04 × 10-7 s-1 for the NO2 and 0.20 vs. 0.24±0.07 × 10-7 s-1 for the O3). Unfortunately, spectral radiometry data are not available for the extensively used bag, which could help to identify whether the reduction in the jNO2 is attributed to the transparency of the walls over usage of the bag or the changes in the production and loss of gases from and to the chamber walls. In the absence of such information, we can only speculate that the changes in the jNO2 over the bag history could be attributable to the differences in the wall loss rates of these gases.*"

13) They further conclude that differences in the way wall losses are estimated and incorporated into data analysis makes a difference in estimates of SOA yield; unfortunately the paper merely shows different results from different data analysis methods without providing any details about how they have applied those methods. Given the work from the same group on the PyCham box model for chambers (O'Meara, …, Shao, McFiggans, Geosci Model Dev., 2021) which seems to incorporate all of the effects that are suggested in this paper, I would have expected a more insightful discussion of the treatment of wall effects that might provide some guidance regarding the best ways to account for these effects, or at least how data from this chamber has been analyzed.

We agree with the reviewer that further discussion is needed to provide adequate descriptions of the techniques and their application methods. Model simulations are now starting to become available and dedicated studies on experimental vs model results will provide insightful information on the wall effects. An extensive model-measurement comparison with methods of constraint of the wall interaction processes is out of the scope of the current paper that aims to describe, for the first time, the MAC and its characterisation. However, we have now included some observations from some first such comparisons for an example experiment, as introduced in the PyCHAM GMD paper referred to by the reviewer. As stated in the response of the comment #5, we are planning dedicated experiments using advanced instrumentation to address the more complex aspects related to vapour interactions, for example. It should be further noted, that according to comments #32, 33, 35 and 36, Sections 3.5, 3.5.1 and 5 have been revised to be more descriptive and easier to follow.

14) The paper also concludes that alpha-pinene SOA yields from the MAC chamber are similar to those determined over the two decades from different chambers, but that differences may be attributed to differences in data analysis. Understanding of the wall effects has advanced significantly in that time, with models similar to PyCHAM playing a key role as reflected in a

number of cited papers. The author thus suggest that regular chamber characterization experiments are needed, and that methods for validating data analysis approaches are needed. This has long been recognized, but bears repeating.

We thank the reviewer for re-emphasising this. Our revised text reflects the importance of such characterisation and consistent analysis.

15) In summary, I find that this paper describing a chamber that has been in use for a decade does not approach the standard of prior presentations and characterizations of new chambers. Nor does the data presented lead to significant new insights.

We have rigorously addressed the reviewer comments and are grateful for their insight in making this a much improved manuscript.

16) Given this system's extensive use, sufficient data almost certainly exists to provide a more rigorous and critical evaluation of the MAC chamber that would allow it to be quantitatively compared to other chambers.

We now include some previous data where it helps address the reviewer comments along with an example of the utility of the interpretation of PyCHAM model data (see response #32). As stated, many of the previous experimental datasets were collected with the aim to understand the particle properties and were less concerned with chamber characterisation, limiting their usefulness for the current manuscript. In recent years, we have become more systematic in addressing repeatability and characterisation of a fuller range of chamber parameters, allowing the studies presented in the paper.

17) The instruments available to the authors could provide data that was not possible at the time that earlier chambers were characterized, so unique data may exist, or certainly could be generated to address the key focus of wall effects. The writing also needs serious work, as does the data presentation.

As it was mentioned in the responses #5 and 13, the "additional instrumentation" is equipment that is not readily available to our group, while addressing the key aspects of the wall effects will require a set of bespoke experiments that is not the main focus of this work. We demonstrate the utility of models such as PyCHAM in moving towards a more complete understanding of chamber behaviour. Here we present the first detailed description and characterisation of the MAC, its comparison with other chambers under a well-established characterisation framework while based on our results, we conclude with some new recommendations for the chamber usage.

The text has changed substantially in the revised version of the manuscript while a thorough check on the language has been conducted. Further, according to the recommendations and the necessary changes in addressing the reviewers' comments, the data representation in the figures has been also changed.

18) In light of my conclusions and to support the authors in their efforts to , I provide additional detailed comments, questions, and points of confusion that came to mind as I read this paper.

Like numerous other chambers, the MAC is fabricated from FEP Teflon. It's volume, 18 m3, is smaller than many but still reasonably large to enable aerosol studies. A feature of the MAC is its interface to an ice cloud chamber, or to sources of gases or particles, including combustion sources.

These design features are indeed key elements of our chamber. The coupling to the cloud chamber, modes of operation and the characterisation of system are presented in Frey et al. (2018) under mixed-phase cloud conditions and are the subject of a manuscript in preparation (Frey et al., 2021) looking at co-condensation of semi-volatile vapours under warm-cloud conditions. The latter includes development and use of the ACPIM model for interpretation of the system performance. We consider the coupling to the cloud chamber as part of an extended facility, beyond scope of the current manuscript and warranting its own extensive characterisation and discussion. Similarly, the coupling to real emission sources, whilst an extremely useful feature, is described in the relevant publications and one further manuscript in preparation. To include all aspects of this into the current paper would extend its scope considerably and necessarily dilute the core material. Nevertheless, these aspects are mentioned in the current manuscript and reference made to the papers where they are discussed.

19) The supporting framework differs from other designs though the description (three pairs of rectangular extruded aluminum frames, with the central rigid pair fixed and the top and bottom pairs free to move vertically. This detail may, or may not be useful to others considering building chambers but the description is uninformative, as is the schematic in Fig. 1. Why is a pair of frames needed at each level? The description in section 2. just mentions three frames. The discussion of the Teflon reactor indicates that it is maintained under slight positive pressure to minimize contamination from lab air; how much pressure is maintained in the chamber?

This links to our response 7, above. The bag is constructed of 4 sections, with joints made between 3 pairs of frames. The clarifying text in response 7 has been included in the revised paper.

Our wording relating to maintenance of a positive pressure was imprecise. The movable counter-weighted frames reduce the possibility of the chamber operating under negative pressure when collapsing on a fixed frame. This aims to minimise instrumental sampling problems and contamination from laboratory air. This description is included in the revision.

20) The chamber illumination differs from many other chambers in being designed to approximately reproduce the atmospheric actinic spectrum in Manchester. The discussion of the illumination system is confusing. It is noted in section 2.3 that the absorption of IR radiation due to water vapour in the atmosphere is simulated by inserting a water-filled container with quartz windows between the light sources and the chamber. What is the purpose of that simulation? It is also suggested that the water is there to minimize heating of the chamber, which is what I would expect is the actual purpose. This should be made clear.

Indeed, our unusual illumination setup enables us to reproduce the atmospheric actinic spectrum in Manchester. We further agree with the reviewer for the purpose of water-filled container is to remove the absorption of IR and thereby the unwanted heat. The quartz plates

are used to reduce the transmission of light from the arc lamps between 305 and 290 nm in wavelength and completely eliminate that below 290 nm.

In order to make the above discussion more clear in the revised version of the manuscript we have altered the wording of the Section 2.3:

*"The irradiation source, consisting of two xenon arc lamps and a bank of halogen bulbs, is mounted inside the enclosure and is used to approximate the atmospheric actinic spectrum. Two 6 kW Xenon arc lamps (XBO 6000 W/HSLA OFR, Osram) are installed on the bottom-left and the top-right of the chamber housing, respectively. Quartz plates with optical polish (PI-KEM Ltd) of 4mm thickness in front of each arc lamp filter out unwanted UV light. The bank of 112 halogen lights, 7 rows of 16 bulbs each (Solux 50W/4700K, Solux MR16, USA), are mounted on the same enclosure wall as the bottom xenon arc lamp, facing the inlet.*

*The unwanted heat generated from the irradiation source is removed by the cooling system which includes the Air Conditioning (AC) unit and a water tank in front of each arc lamp with circulating water system. The chiller water circulates running through aluminium bars cooling the halogen bulb holders and through tanks in front of each arc lamp faced by the quartz filter plates, in order to dissipating heat produced by absorption of unwanted IR light by water vapour. "*

21) The air supplied to the chamber is, according to section 2.4, dried lab air that is passed through a series of packed bed scrubbers, two to remove gases, and a HEPA filter. Figure 1 shows the air being drawn from the chamber, suggesting that chamber air is recirculated. Which is it? How pure is the processed air? This could be documented in terms of measured contaminant levels.

The chamber air is not recirculated, the inlet airflow and outlet airflow each only flow in one direction through the stainless steel pipework. A legend has been added to Figure 1 of the original manuscript to make the direction of airflow clearer in filling and flushing the chamber (shown as Fig. 4 below). The change in modes is effected by positioning of three 3-way and one 2-way electro-pneumatic valves, though the blower operates in one direction throughout.

In terms of the purity of our processed air, as can be seen from our response to the comment #8, along with the associated figures, after filling and flushing MAC several times we can reach to concentrations of NOx<10ppb, O3<2ppb, particle number <20 particles cm-3 and particle mass of 0 $\mu$g m$^{-3}$ that implies that our processed air is at least of that purity.

[Figure]

**Figure 4:** Updated schematic of the MAC

22) A custom-built humidifier is included, but its description provides no hint as to how the water vapor is introduced into the air, nor is any indication as to the purity of the water used indicated. Contaminants in the water could introduce particles, depending on the humidification method.

As it was stated in the L156-157 of the original manuscript, the water that was used to fill the humidifier was ultra-pure (UP; i.e., resistivity of ≥18.2 MΩ-cm). The UP water was heated to ~80°C using an immersion heater, thereby producing water vapour.

In order to include this information, the related sentence has been modified as:

*"The custom-built humidifier comprises a 50L tank fed with ultra-pure water (resistivity ≥18.2 MΩ-cm), producing water vapour using an immersion heater that heats the water to ~80°C."*

As shown in aforementioned Fig. 4 (Fig. 1 in the revised manuscript), the purified lab air that is used to fill the chamber can be directed through the humidifier, the ozoniser and aerosol residence chamber and carry any of their components to the chamber while filling at high flow rate. In order to include the above information, a sentence has been added at the end of Section 2.4 that reads as:

*"As shown in Fig. 1, the purified lab air that is used to fill the chamber can be directed through the humidifier, the ozoniser and aerosol residence chamber and carry any of their components to the chamber while filling at high flow rate (3 m3 min-1), ensuring rapid mixing (see Section 3.2)."*

23) Seed aerosols are generated using an atomizer, but there is no mention of neutralizing the excess charge on the droplets. If as-sprayed particles are introduced to the chamber without neutralizing them first, charge induced wall losses could be substantial. Charan et al. (Cited) discuss the effects of particle charge on losses, though not the danger of not neutralizing the aerosol.

Seed aerosol that were generated in our chamber were not neutralised and charge-induced wall losses could hypothetically be substantial in our chamber. However, all of our characterisation experiments were conducted under the same conditions and therefore such losses were reflected to the measured size-resolved loss rates of the particles. Moreover, our comparison of measured wall-losses in inorganic seeded experiments using an atomiser are broadly consistent with the size-dependent losses required to tune PyCHAM model-measurement agreement in nucleation experiments (see response #32).

24) The MAC facility includes a control system to automate the filling and cleaning of the chamber. This could enhance reproducibility from experiment to experiment. As other systems do not have such capability, this feature of the system being reported might warrant further discussion, including data that shows the extent to which experimental conditions can be duplicated.

We agree with the reviewer that our manuscript need to provide experimental evidence to support the reproducibility of experiment that conducted by the MAC. As already shown in the responses of the comments #8, 10 and 11, evidence is now provided regarding the reproducibility of our experiments both in terms of background levels as well as the repeatability in our injections and thereby initial concentration of reactants across a number of experiments.

25) The discussion of the instrumentation used on the chamber (section 2.8) describes the "core instruments." Some of the additional instruments listed in Table 1 are mentioned.

As mentioned in response #5, the sub-list of the "additional instrumentation" correspond to instruments that are shared with our broader Centre for Atmospheric Science (CAS) group at the University Manchester and are not routinely available outside dedicated projects. To make this more clear, we have altered the sentence found in the L226 of the original manuscript to read as:

"*A selection of additional instrumentation that are shared within the Centre for Atmospheric Sciences (CAS) group at the University of Manchester, are potentially available to be used on demand.*"

26) Since data presented in the paper include transient aerosol wall loss measurements, it is interesting that the DMA in the core set is used as a DMPS, in which the voltage is stepped. Such measurements tend to be slow due to the time required to attain steady state after each voltage step. The information provided is insufficient to assess whether this is an issue. One of the unmentioned additional instruments is an SMPS that might provide better time response.

As mentioned in the response of the comments #5 and 25, the "additional instrumentation" listed on Table 1, where the SMPS is found, are instruments that are not routinely available to our experiments. The DMPS does indeed have a lower time-resolution than the SMPS, i.e., 10 vs. 2 mins, respectively. Nonetheless, as it was shown in previous works (e.g., Hamilton et al., 2011; O'Meara et al., 2021), the time-resolution of the DMPS can adequately capture the evolution of the SOA particle size formed in our chamber under our typical experimental conditions. Consequently, the lower time-resolution of the DMPS compared to a SMPS has

thitherto not prevented us from describing the size evolution of the particles in the MAC for the purposes of our studies. Additionally, scanning inversion algorithms can risk artificial distribution broadening and skewing that is avoided by stepping where time resolution is not a concern.

27) This point will come up again in the discussion of the experimental results. A filter on the outlet flow from the chamber at the end of the run. Figure 1 suggests that the filter processes both inlet and outlet flows.It should be made clear which flow is sampled. Moreover, I presume that there is a bypass flow, and valves to select whether the filter is used. That should also be made clear.

We have added the legend in figure 1 to explain the inlet airflow and outlet airflow. The filter is only collected on the outlet airflow from the chamber at end of each experiment. The plane of the filter holder is shown in the updated schematic the MAC (Fig. 4 in response #21) and air is only collected in the outlet flow from the final flush after each experiment. We have adjusted the text in the manuscript to be clearer.

28) Section 3 of the paper presents chamber characterization data. Figure 2 shows that the temperature is affected by the lights, and that the temperature at the center of the chamber is higher than that of the wall. Interestingly, the relative humidity is also higher at the center than at the wall. In a well mixed chamber, one would expect the RH to decrease as T increases. For the RH data, two different sensors are used. Is the trend an artifact of the sensors, perhaps due to the sensors being radiatively heated? Some discussion of these results is needed.

We agree that some additional discussion of these results might be needed. Indeed, as the reviewer points out, the differences could likely be attributed to the radiative heating. This is particularly evident in the light experiment opposed to the one conducted in the dark, where the RH and T difference between the centre and the edge of the bag is larger. In contrast, in the dark experiments, the RH and T differences between the centre and edge of our chamber are falling within the measured variance, i.e., $40 \pm 1\%$ vs. $39 \pm 1\%$ RH and $24 \pm 1°C$ vs $23 \pm 1°C$ T.

To reflect this, in the revised manuscript we added some relevant discussion that reads as:

"*Figure 4 shows the temperature and relative humidity, measured at the edge (Edgetech sensor) and at the middle of the MAC (Sensirion sensor) for three characteristic experiments; one conducted in the dark and two in the presence of light. In the light experiments, it appears that both the temperature and humidity were higher in the centre of the MAC than that of the wall, while in the dark experiments these differences were negligible as they were within the uncertainty of our measurement. A likely explanation for this unexpected behaviour in the light experiments can be possibly to the radiative heating of the sensors in these experiments that could result in an over-estimation of the RH.*"

29) Mixing is explored with both gas and particle tracers. Both approach a reasonably steady level after a few minutes. The discussion of the particle experiments indicates that neutral seed particles were introduced into the chamber. Were the particles actually neutralized?

The particles are not neutralized in that each particle may retain excess charge, but neutral was used to refer to them comprising neither basic nor acidic electrolyte solutions. Specifically, we

generated particles by atomising aqueous solution of $(NH_4)_2SO_4$ instead of $NH_4HSO_4$. To make it clear, we changed it to non-acidic in the paper.

30) The paper states that the light intensity as 3.5 times less intense than, but closely approximates that of the atmospheric actinic spectrum, but the lamps do not uniformly illuminate the chamber. Two different types of lamps were used to reproduce the spectrum, but they do not appear (from Fig 1) to illuminate the same portion of the chamber volume. How was the spectrum measured to produce the results shown in Fig. 4?

The total actinic flux was measured at the middle of the bag, roughly 150 cm away from each of the arc lamps in both vertical and horizontal direction. As we further mention in the Section 2.1 of the original manuscript all the enclosure of the chamber is covered in "space blanket" that can assist to a more uniform distribution of the light in the MAC. To provide additional information about our actinic flux measurements we added the following text in the Section 3.3 that reads as:

*"The artificial radiation in the MAC has a broad radiation distribution owing to the chosen combination of illumination sources, producing irradiation over the wavelength range 290-800 nm to capture all wavelengths of the atmospheric actinic spectrum. Figure 6 shows the total actinic flux measured in MAC (red line) multiplied by 3.5 compared with the Manchester midday clear sky measurements on a June day. The total actinic flux in MAC was measured the centre position of chamber bag (150cm apart from the arc lamps in both vertical and horizontal axes)."*

31) The authors report three different photolysis rates; their respective deviations from atmospheric rates differ substantially. Since considerable effort was invested in matching the actinic spectrum, one might expect some discussion of the reasons for the differences.

We agree that additional discussion could be beneficial. The photolysis rate of $NO_2$ ($jNO_2$) as derived from our steady state actinometry experiments was comparable, within our measured variability, with that directly measured from the integrated absorption across the measured wavelengths ($2.25 \pm 0.4$ vs. $1.5 \times 10^{-3}$ s$^{-1}$, respectively). Given that the $jNO_2$ obtained by the actinometry experiments is an average and is estimated based on the assumption of photostationary state for trace gases in the bag, while the spectral radiometry is a point-measurement in an imperfect integrating sphere, which could not be representative for the whole chamber, these results are in a reasonable agreement. The integrated $jNO_2$ measured by spectral radiometry in the ambient Manchester on clear sky over the summer was $7 \times 10^{-3}$ s$^{-1}$ but had comparable light spectrum to that measured in MAC. The values obtained in MAC are more similar to those reported previously over the winter-time at Finokalia station, Greece and are generally comparable with those obtained across the broader simulation chamber community, as shown in Table 2 of the original manuscript.

In order to reflect this, additional discussion has been added in the revised manuscript that reads as:

*"In the MAC, the photolysis rate of $NO_2$ ($jNO_2$) as derived from our steady state actinometry experiments was comparable, within our measured variability, with that directly measured from the integrated absorption across the measured wavelengths ($2.25 \pm 0.4$ vs. $1.5 \times 10^{-3}$ s$^{-1}$, respectively). Given that the $jNO_2$ obtained by the actinometry experiments is an average and is estimated based on the assumption of photostationary state for trace gases in the bag, while*

*the spectral radiometry is a point-measurement in an imperfect integrating sphere, which could not be representative for the whole chamber, these results are in a reasonable agreement. The integrated jNO₂ measured by spectral radiometry in the ambient Manchester on clear sky over the summer was $7 \times 10^{-3} s^{-1}$ but had comparable light spectrum to that measured in MAC. The values obtained in MAC are more similar to those reported previously over the winter-time at Finokalia station, Greece (Gerasopoulos et al., 2012) and are generally comparable with those obtained across the broader simulation chamber community, as shown in Table 2."*

32) The discussion of aerosol wall losses is particularly disjoint. Wall losses were shown in Crump et al. (1983 — not 2007) to depend strongly on particle size. The model from Crump, as extended to include charge effects by McMurry, is widely used to correlate observed chamber wall losses with turbulence levels within the chamber in order to properly account for losses in inferring aerosol dynamics from chamber data. Fig. 5 presents size dependent wall loss rates at two different relative humidities, and two different turbulence levels, i.e., that when the air conditioner is disturbing the chamber walls and that with the air conditioner turned off. The data, apparently from a number of different experiments, show substantial scatter. Since no mention was made of charge conditioning (neutralizing) the seed aerosol, I must ask whether the scatter in the data could be due to charge effects. This is suggested without any elaboration in section 3.5. Discussion of these effects is needed. It would be good to compare the observed size dependent losses to the Crump model, and to assess whether the wall loss parameters change between experiments in each class (RH and AC).

We acknowledge that the contents of Section 3.5 might needed some further elaboration. It is absolutely correct that the particle wall losses are size-dependent. Whilst Crump et al., (1983) model might be able to simulate the losses due various turbulence levels, it was designed for a cylindrical continuous flow reactor opposed to our batch mode chamber, where the flow dynamics are more complex than the model.

Our experimental findings suggest that the active mixing (effected with the air circulation in the chamber enclosure by the AC) and the relative humidity levels can affect the particle decay rates). The higher loss rates when the active mixing was on can be possibly attributed to enhanced turbulent deposition, while the differences between the low and high RH levels were relatively small and within the variability of the measurements.

The large variability in the size-resolved particle wall loss rates within each of the selected experimental conditions can be possibly attributed to the changes in the chamber behaviour, considering that the experiments averaged here were conducted sporadically over a large time-period (2017-2019) and on different conditions of the bag. We discuss this further in Section 3.5.1 (please see also response to comment #35 below). Alternatively, indeed, this scattering could be also hypothetically attributed to particle and/or chamber charging effects (Charan et al., 2019), considering that our seed aerosols were not neutralised and the experiments were conducted over a large timeframe; hence making difficult to assess experimentally any potential chamber charging effects.

However, the combination of our experimental results presented in this study with those presented earlier using our newly developed PyCHAM model (O'Meara et al., 2021) can provide some further insights on the latter manner. Figure 5 below shows the measured size-resolved particle decay from a number of identical wall-loss characterisation experiments (i.e.,

ammonium sulfate seed in the dark), conducted over a span of 3 years and at various conditions of the bag. Additionally, the size-resolved wall-loss rates that were required to reproduce the SOA formation in the limonene nucleation experiment presented in the O'Meara et al., (2021) study are also shown. Evidently, the variation in the measured loss rates of the particles nebulised in similar manner in the MAC as a function of the chamber bag history can be substantial. Interestingly, the modelled particle losses that required to reproduce a nucleation experiment, where no induced charge of the particles is expected, are comparable with those measured in a new bag. What is more, the model of McMurry and Rader (1985) suggests that the differences in the wall-loss rates of the particles having 0 and +1 charge can be as a high as 2 orders of magnitudes (or more) for particles of 100 nm in diameter, while the differences between +1 and +3 charges were modest (see Fig. 9 on O'Meara et al., 2021). Here, the observed differences between the potentially charged AS particles in the characterisation experiments were within the same order of magnitude to those modelled for a nucleation experiment, where no particle charge is expected. Therefore, this analysis demonstrates that neutralising the seed aerosols prior the injection to the MAC likely has less of an effect than the history of the bag.

In order to include the above, Section 3.5 was altered and now it reads as:

[revised manuscript text omitted]

33) The paper also discusses particle number and mass loss rates; since these are averaged over the measured size distributions, they are expected to differ from one experiment to another, due to differences in the size distribution, and within any experiment as the size distribution changes with time. Those values, and the discussion of them convey little useful information and distract from and confuse the paper. The data presented in Table 3 provides a qualitative estimation that losses in the MAC are comparable in order of magnitude to those in other chambers. The comparison with the CMU chambers in terms of the size-resolved decay rate are much more fundamental and significant.

We agree with reviewer's suggestion. Indeed, the particle losses are largely size-dependent therefore expressing the wall loss rates as average values might not be an insightful finding for a characterisation paper. We have removed the discussion regarding the average number and mass loss rates and we have focused the section in providing discussion for the size- resolved

losses only. Please see the revised Section 3.5 that can be found also copied in the response of the comment #32.

34) Mixing and boundary layer effects also affect losses of gaseous species. Looking at individual species eliminates the confounding effect that size distribution introduces for particle losses, but they are confounded by differences in deposits on the walls into which the vapors may partition.

We thank the reviewer for their valuable insight, which we already appreciate. We are not sure what the reviewer is suggesting we do here.

35) Section 3.5.1 presents a very confusing discussion of the effect of different wall-loss correction methods on estimates of SOA formation, using the mass loss rate in addition to the size-dependent loss rate. The mass loss rate is irrelevant as noted above. The Verheggen model treats the eddy diffusion and turbulent kinetic energy as empirical parameters based upon the Crump model, and uses the general dynamic equation to estimate the value of the relevant parameters. The large difference between the size resolved data inversion and the Verheggen method therefore requires some discussion. Experiments performed with seed aerosol alone provide a basis for comparison between the two models. I can only speculate as to why the Crump model gives consistently higher values than the Verheggen model which clearly includes coagulation and other aerosol dynamics processes. Since there is no discussion as to how the data were inverted it is unclear what processes were included in the data analysis.

As also stated in the response of the comment #33, we agree with the reviewer that the total particle loss rates might not expressing adequately the particle losses that are size-dependent. Therefore, we have removed this approach from that Section and related figures.

The differences between the size-resolved correction approach and the Verheggen model are likely partly attributable to coagulation. Additional discussion regarding the inversion methods and analysis procedures has been included in the revised manuscript that reads as:

"*The different approaches clearly result in substantially different wall-loss corrected SOA masses (Fig. 9). In all cases, the correction using the ammonium sulfate size-resolved wall-loss rates resulted in greater differences compared to the Verheggen and Mozurkewich (2006) model. These differences can be, at least partly, attributed to the parameters accounted in each method. The size-resolved particle correction applies the measured particle decay rates from the characterisation experiments to the decay of the particles in the SOA experiments. Effectively, in this method it is assumed that the losses of the AS particles in the characterisation experiments (from any loss process) are the same as those formed in the SOA experiments. On the other hand, the Verheggen and Mozurkewich (2006) model employs inverse modelling to simulate the particle wall losses based on diffusion and gravitational settling, while the losses due to coagulation are indirectly inferred and the eddy diffusion and the turbulent kinetic energy are treated as empirical parameters based on the Crump et al., (1983) model. Therefore, the differences between the two approaches could be partly attributed to the particle losses due to coagulation that is indirectly accounted in the Verheggen and Mozurkewich (2006) model opposed to the size-resolved correction. Clearly, treating the particle losses to atmospheric simulation chambers is not a trivial task and this could have substantial impacts for the reported SOA yields.*"

As we only measure the size-resolved particle decay rate in the characterisation experiments, correcting the data of that particular seed-only experiment will inevitably result in a flat line, so it will not be possible to get insightful information about the different wall-loss correction methods by looking at seed-only experiments. More importantly, the measured size-resolved particle decay rates from characterisation experiments, such as those described here, are widely used from the chamber community to correct the SOA mass formed and thereby report SOA particle yields from various precursors. Therefore, we believe that more meaningful information can be gained when comparing the different wall-loss correction methods to real experiments.

It should be noted here that in the light of the responses of the following comment (see responses to comment #32 and 36), we have simplified our approach and we have removed the "average from all experiments" corrections and we are now showing the effect of the wall-loss vs aging of the bag using a different approach.

36) The discussion of SOA formation analysis is further confused by using both data from the "nearest" experiment and the average of all experiments. I would be much more comfortable with an analysis of a clearly defined experiment. Teflon chambers age with use. A systematic look at the individual seed aerosol loss experiments should provide direct evidence for that.

We agree with the reviewer that the discussion and the presentation of the results regarding the effect of the bag usage to the wall loss rates may have been confusing. Consequently, we have removed from Fig. 6 the corrections made from the "average from all experiments" and we now show only the corrections from using the "nearest" experiment (for further details please see response to the comment #35). As suggested by the reviewer, we have created a new figure (Fig. 8 in the revised manuscript and shown here as Fig. 5 in the response #32), where we show the size-resolved particle decay rates (s$^{-1}$) in experiments conducted in used and clean bag and we have added a relevant discussion in Section 5 that reads as:

"*In addition to the changes in the decay rates of gases, similar changes were observed in the wall losses of particles. Figure 8 shows the measured size-dependent particle decay rates (s-1) in characterisation experiments conducted at different conditions of the bag. More specifically, characterisation experiments conducted in an extensively used bag after a campaign using diesel engine fumes, in a used bag after a campaign on SOA formation and in a new bag are shown. Clearly, the size-dependent losses of the particles can be substantially affected by the condition and the usage of the bag. Wang et al., (2018) reported significant changes in the wall-loss rates of particles after major maintenance activities in the area where the chamber was suspended and attributed those differences to the electrostatic forces caused by friction. In our setup, the chamber is suspended and enclosed in a housing and the operators have little to no contact with its walls, so it may be unlikely for this to be the main cause of changes in the particle wall-losses over the bag usage history. Considering that the correction of the SOA mass and particle yield calculations are strongly dependent on the measured particle loss rates in characterisation experiments, at least for MAC, it is recommended to conduct more frequent particle and gas loss characterisation experiments to enable more reliable corrections.*"

37) Section 3.6 focuses on wall reactivity, which is a very real factor in data analysis (Huang et al., Env. Sci. Tech., 2018). The section should be labeled something like Chamber Wall Reactivity since calling this "Auxilliary Mechanism" conveys no useful mechanism.

We agree with the reviewer and had changed the section name in the manuscript

38) Huang et al. develop a model for the wall reactions that could provide a basis for considering the processes that are very briefly discussed in this section.

Indeed, gas phase molecules can be lost to the boundary layer of the surface chamber wall by molecular diffusion and macroscopic mixing, while their reactive uptake by the Teflon film and any deposited material is also possible. Teflon film can act as a reservoir for organic vapour deposition during chamber experiments that may contribute to $O_3$ loss by oxidation. Furthermore, the organic compounds deposited can act as absorptive mass, in turn influencing the mass transfer from the gas phase to the walls (Charan et al., 2019). These are all processes that can affect the wall interactions and are discussed in the revised version of the manuscript as:

"*Gas phase molecules can be lost to the boundary layer of the surface chamber wall by molecular diffusion and macroscopic mixing, while their reactive uptake by the Teflon film and any deposited material is also possible. Teflon film can act as a reservoir for organic vapour deposition during chamber experiments that may contribute to $O_3$ loss by oxidation. Furthermore, the organic compounds deposited can act as absorptive mass, in turn influencing the mass transfer from the gas phase to the walls (Charan et al., 2019).*"

39) The discussion of effects of wall contamination addresses problems that are encountered in every chamber that is used form multiple experiments between baking or other rigorous cleaning (which are not possible with teflon chambers. The data provided give some indications of the changes in performance of the chamber with age.

We thank the reviewer for acknowledging the novelty of the work that aimed to provide some further information in this manner based on our available measurements.

Specific points:

The abstract and introduction should make it clear that the chamber described in this paper has been used for over a decade, since papers of this sort usually report on relatively new experimental systems.

We thank the reviewer for this suggestion. This has been clearly described in the updated manuscript.

L 21: Provide a meaningful name rather than "auxilliary mechanism."

The "auxilliary mechanism" has been replaced by "chamber wall reactivity" as suggested.

L. 65: It would also be useful to include the CLOUD chamber at CERN in the list of rigid chambers that can simulate the free troposphere.

Added.

L. 68: The discussion confuses the Caltech dual chambers (both ~28 m^3) of Cocker et al. (2001) and the University of California at Riverside dual chambers (90 m^3) (Carter et al., 2005).

The sentence has been rephrased to:

"... Caltech dual chambers (both 28 m$^3$) (Cocker et al., 2001a), the University of California at Riverside dual chambers (both 90 m$^3$) (Carter et al., 2005)..."

L. 73: that can act as a sink

Addressed.

L 74: and aerosol particles (), and as a surface

Added.

L. 79: While it is obvious to those who work with chambers that chambers introduce background and memory effects, some explanation is needed for those with less experience or who may be reading this to understand data from experiments on this chamber.

The sentence has been rephrased to:

"In addition, gases and particles as well as the intermediate reactants can interact with and partition into chamber walls (so-called memory effect), which can affect repeatability and reliability of the results"

L. 81: The sentence beginning "This necessitates …" is ambiguous. This what?

Clarified. "This artifact due to the memory effect..."

L. 83: MAC is used for the first time in the body of the paper. Spell it out.

Added.

L. 152: Is the specific blower type needed. There's a lot in this sentence, and inclusion of the blower details makes it difficult to follow.

The blower is not specific type. The details might make it a bit difficult but will be better to add the instrument model details as consistent with the remaining of the manuscript.

L. 158: using ECD grade — and always spell out abbreviations when they are introduced for the first time.

Addressed.

L. 176: Edgetech and. Sensirion sensors (plural)

Addressed.

L. 219: The DMA uses filtered chamber air

Addressed.

L. 233: The company name is Cambustion - with an A

Addressed.

L. 235: Droplet Measurement Technologies. Spell out the name properly/

Addressed.

L. 239: additional instruments …

Addressed.

L. 250: the chamber temperature is controlled by the air conditioning system; that of the chamber itself is measured…

This sentence has been rephrased to

” The temperature in MAC is controlled by the AC system, which compensates for the heat generated by the illumination system”

L 323: and O3 at concentrations ranging from 230 ppb to 350 ppb

Rephrased.

L. 332: GIG-CAS, while the

Addressed.

L. 335: Measured first-order wall loss rates of selected

Addressed.

L. 344: Crump et al. 1983 — also correct the citation in the bibliography.

Addressed.

L. 381: Investigation of various particle wall-loss

Addressed.

L. 429:  Eq. 2 should be inserted after this line

Addressed.

L. 433:  Eq. 3 should be inserted after this line

Addressed.

L. 475:  You really don't mean that the photolysis rates differ by 3 orders of magnitude.  State the numbers independently and correctly.

The numbers are changed to "...(1.83 ± 0.47 × $10^{-3}$ vs. 2.25 ± 0.40 × $10^{-3}$ s$^{-1}$, respectively)...".

L. 477: A substantial increase in the wall …and O3 was observed

Addressed.

L. 505: Note that the air circulation around the chamber also enhances wall losses.

Added.

L. 515:  Mixing also affects losses of gases

Added.

L. 519:  Differences between particle mass and number losses depend on the size distribution and are, therefore, not meaningful.

Deleted.

L. 535: as shown in Fig. 7

Addressed.

L. 538: e.g., gases … — comma is needed

Addressed.

**Citation**: https://doi.org/10.5194/amt-2021-147-RC2